

# Closed-channel parameters of Feshbach resonances

**Pascal Naidon**

Few-Body Systems Physics Laboratory, RIKEN Nishina Centre, RIKEN, Wakō, 351-0198 Japan

pascal@riken.jp

## Abstract

This work investigates how the closed channel of a Feshbach resonance is characterised by experimental observables. Surprisingly, it is found that the two-body observables associated with the Feshbach resonance can be insensitive to the properties of the closed channel. In particular, it is impossible in this situation to determine the energy of the bound state causing the resonance from the usual experimental data. This is the case for all magnetic Feshbach resonances in ultracold atoms, due to their deep two-body interaction potentials. This insensitivity highlights a major difference with Feshbach resonances that involve shallow interaction potentials, such as hadron resonances. It appears however that short-range two-body correlations and three-body observables are affected by a parameter of the closed channel called the "closed-channel scattering length". A photoassociation experiment is proposed to measure this parameter in ultracold atom systems.

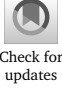

# 1  Introduction

Feshbach resonances [1,2] are found in many quantum systems, occurring whenever a continuum of states couples to a bound state. They are particularly important in the field of ultracold atoms, where the "magnetic Feshbach resonances" [3,4] have provided the possibility to control interatomic interactions through the application of a magnetic field [5,6]. The concept of Feshbach resonance is also used in hadron physics to account for exotic bound states or resonances close to hadron thresholds [7–9], and its relevance to condensed matter systems has recently been pointed out [10,11].

   Many of the previous theoretical studies of Feshbach resonances have been concerned with building up models that reproduce experimental data [12–18]. In the present work, an opposite approach is taken by considering which parts of the model are constrained by the observables. For this purpose, the two-channel model describing Feshbach resonances is introduced in Sec. 2, followed in Sec. 3 by a generic example showing an explicit dependence of observables on the properties of the bound state responsible for the resonance. Then, the regime where this dependence disappears is presented in Sec. 4, followed by a discussion of some remarkable aspects of this regime. Finally, the possibility to probe the closed-channel properties from short-range physics is examined in Sec. 6.

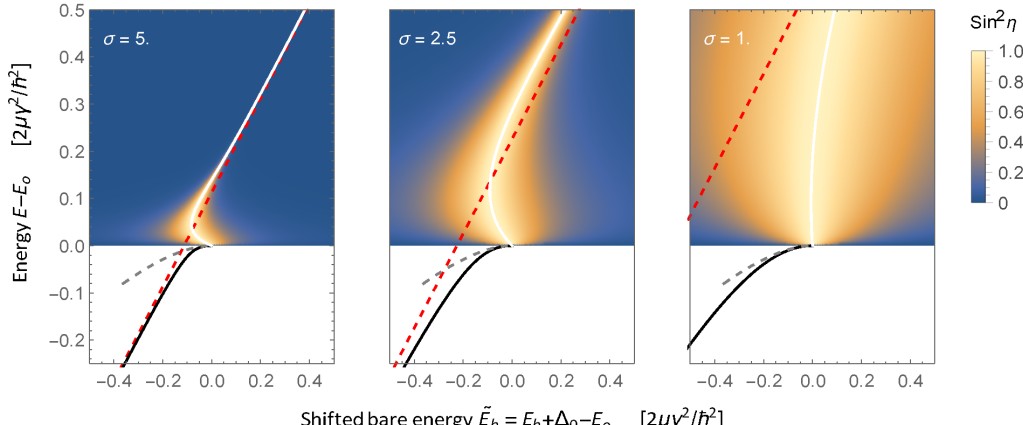

Figure 1: Energy spectrum and scattering phase shift of the non-relativistic Gaussian model of Feshbach resonance as a function of the energy difference $\tilde{E}_b$ between the shifted bare energy $E_b + \Delta_0$ and the threshold energy $E_o$, for a fixed reduced width $\gamma$. The results for three different values of the closed-channel parameter $\sigma$ (in units of $\hbar^2/2\mu\gamma$) are shown in the left ($\sigma = 5$), middle ($\sigma = 2.5$), and right ($\sigma = 1$) panels. In each panel, the shading above the continuum threshold shows the quantity $\sin^2\eta$ obtained from Eqs. (2) and (5), and the solid black curve below the continuum threshold shows the dressed bound state energy given by Eq. (6). The resonance ($\sin^2\eta = 1$) is shown as a white curve. The dashed grey curve shows the dressed energy in the limit $\sigma \to 0$, which coincides with the QDT Eq. (8) in the zero-range limit given by Eq. (11). The dashed red line shows the energy of the bare bound state in the closed channel causing the resonance. One can see that the value of $\sigma$ significantly affects the spectrum and scattering, making it possible to determine $\sigma$, and thus the bare bound state energy, from these observables.

## 2 Two-channel model

The following analysis is restricted to isolated resonances of a two-particle system, i.e. a single two-body bound state $|\phi_b\rangle$ coupled to a two-body continuum. More specifically, the bound state $|\phi_b\rangle$, which is called the "bare bound state", is assumed to occur in a "closed channel" described by a Hamiltonian $H_{cc}$, such that $(H_{cc} - E_b)|\phi_b\rangle = 0$, and this closed channel is coupled through coupling terms $H_{oc}$ and $H_{co} = H_{oc}^\dagger$ to an "open channel" described by a Hamiltonian $H_{oo}$ featuring a scattering continuum above a certain threshold $E_o$. The two channels correspond to two different internal states of the particles, such as two hyperfine states of two atoms, or two quark configurations of a hadron. The isolated resonance theory of this two-channel model shows that the system at energy $E$ is described by the complex energy shift (see Appendices A and B)

$$\Delta^+(E) \equiv \langle \phi_b | H_{co} | (E + i0^+ - H_{oo})^{-1} | H_{oc} | \phi_b \rangle, \tag{1}$$

whose real and imaginary parts $\Delta$ and $-\Gamma/2$ define respectively the shift and width of the resonance.

For energies $E$ above the open-channel threshold $E_o$, the scattering properties are strongly modified for energies around the energy $E_b$ of the bare bound state. Indeed, in a certain partial wave set by the angular momentum of $\phi_b$, the scattering phase shift,

$$\eta(E) = \eta_{\text{bg}}(E) - \arctan \frac{\Gamma(E)/2}{E - E_b - \Delta(E)}, \tag{2}$$

can reach unitarity (i.e. $\sin^2 \eta = 1$) at a particular energy, corresponding to a resonant state. Here, $\eta_{\mathrm{bg}}$ denotes the "background" scattering phase shift away from that resonance. In the following, the s wave will be considered, although other partial waves can be treated in the same way. In this case, in the limit of small scattering wave number $k \equiv \sqrt{2\mu(E-E_{\mathrm{o}})}/\hbar$ (with $\mu$ being the reduced mass of the two scattering particles), the scattering properties are governed by the s-wave scattering length $a \equiv -\lim_{k\to 0} \eta/k$. From Eq. (2) one finds

$$a = a_{\mathrm{bg}} - \gamma / (E_{\mathrm{b}} + \Delta_0 - E_{\mathrm{o}}) \,, \tag{3}$$

where $a_{\mathrm{bg}} \equiv -\lim_{k\to 0} \eta_{\mathrm{bg}}/k$ is the background scattering length away from resonance, $\gamma \equiv \lim_{k\to 0} \Gamma/2k \geq 0$ will be referred to as the "reduced width" [19], and $\Delta_0 \equiv \lim_{k\to 0} \Delta(E)$ is the zero-energy shift. Equation (3) shows that the scattering length can be arbitrarily large when the bare bound state energy $E_{\mathrm{b}}$ shifted by $\Delta_0$ approaches the threshold $E_{\mathrm{o}}$. This divergent behaviour of the scattering length is the basis for its control in ultracold-atomic systems by tuning $E_{\mathrm{b}}$, thanks to its dependence on an applied magnetic field.

For energies $E$ below the open-channel threshold $E_{\mathrm{o}}$, the coupled system may feature a "dressed bound state" (called "Feshbach molecule" [3, 20] in the context of ultracold-atom physics) whose energy $E_{\mathrm{d}}$ is shifted from the bare energy $E_{\mathrm{b}}$ according to the formula:

$$E_{\mathrm{d}} = E_{\mathrm{b}} + \Delta(E_{\mathrm{d}}) \,. \tag{4}$$

Whether the effect of the bare bound state appears as a resonant state above threshold, or a dressed bound state below threshold, or both, depends on the value of the shifted energy $E_{\mathrm{b}} + \Delta_0$ with respect to the threshold $E_{\mathrm{o}}$. It is readily seen from Eq. (1) that both the resonant and dressed bound state will in general depend on the characteristics of the bare bound state $|\phi_{\mathrm{b}}\rangle$ and the coupling $H_{\mathrm{co}} = H_{\mathrm{oc}}^\dagger$. These characteristics thus introduce "closed-channel parameters" [21] into the problem. Let us now investigate how these parameters affect observables.

## 3 Generic example

A simple example is shown in Fig. 1 corresponding to a well-known non-relativistic model [22–24] where there is no interaction between particles in the open channel, and the coupling factor $\langle \boldsymbol{k}|H_{\mathrm{oc}}|\phi_{\mathrm{b}}\rangle$ is taken to be of the isotropic Gaussian type $W_0 \exp(-k^2\sigma^2/2)$, where $W_0$ and $\sigma$ constitute here the closed-channel parameters. In this case, the shift and width above threshold can be calculated analytically:

$$\Delta(k) = \Delta_0 + \frac{\Gamma(k)}{2}\mathrm{Im}\bigl[\mathrm{erf}(ik\sigma)\bigr], \qquad \frac{\Gamma(k)}{2} = \gamma k e^{-k^2\sigma^2} \,, \tag{5}$$

as well as the dressed energy $E_{\mathrm{d}}$ below threshold:

$$E_{\mathrm{d}} = E_{\mathrm{b}} + \Delta_0 + \gamma\kappa e^{\kappa^2\sigma^2}\bigl(1 - \mathrm{erf}(\kappa\sigma)\bigr) \,, \tag{6}$$

where erf is the error function and $\kappa = \sqrt{2\mu(E_{\mathrm{o}} - E_{\mathrm{d}})}/\hbar$ is the binding wave number. In this model, the reduced width is given by $\gamma = \frac{2\mu}{4\pi\hbar^2}W_0^2$, and the zero-energy shift by $\Delta_0 = -\frac{\gamma}{\sqrt{\pi}\sigma}$. Assume that the scattering phase shift can be measured for different scattering energies (as in high-energy experiments) or the dressed bound state energy can be measured for different values of $E_{\mathrm{b}} - E_{\mathrm{o}}$ (as in ultracold-atom experiments). Then, fitting the data by Eqs. (5) or Eq. (6) should in general unambiguously determine the parameters $\gamma$, $\sigma$ and $E_{\mathrm{b}} - E_{\mathrm{o}}$.

Figure 1 illustrates how different values of the closed-channel parameter $\sigma$ at fixed $\gamma$ lead to different scattering phase shifts $\eta(E)$ and different dressed bound-state energies $E_{\rm d}$. For this particular model, only three different measurements are required to determine the three parameters of the model, enabling a characterisation of the bare bound state.

This Gaussian model can be regarded as the regularised version of a contact model corresponding to the leading order of the low-energy effective field theory describing the resonance, as is often done in the context of nuclear [25] or hadron resonances [26]. After renormalisation, the parameter $\sigma$ can be set to arbitrarily small values to recover the contact limit, yielding results that are independent of $\sigma$. This $\sigma$-independent universal theory is valid in a low-energy region (i.e. close to the threshold), which can be seen in Fig. 1 where the dressed bound state energy curves for different values of $\sigma$ all coincide with a universal curve shown in dashed grey — we come back to this point in Sec. 4.3. Away from this region, the effective field theory requires higher orders, which, like the simple Gaussian model with finite $\sigma$, introduce parameters characterising the closed channel. Again, the general conclusion holds in this case: with enough experimental data, these closed-channel parameters can in principle be determined.

# 4 Quantum defect theory regime

## 4.1 Insensitivity to the closed channel

It will now be shown that there is a regime where the details of the closed channel are undetermined by experimental observations of the scattering shift or binding energy. This situation arises for systems in which the inter-channel coupling occurs around a distance $r_{\rm w}$ where the open-channel wave functions are energy independent. This happens when the potential $V_{\rm o}(r)$ in the open channel has the form $E_{\rm o} + V_{\rm tail}(r)$ beyond a certain distance $r_0$, where $V_{\rm tail}(r) \xrightarrow[r\to\infty]{} 0$ is a potential tail that is independent of the value of the open-channel scattering length $a_{\rm o}$, which is set by the form of $V_{\rm o}(r)$ at shorter distances $r \lesssim r_0$. If the tail is deep enough, for a given energy $E$, there is a range of distances $r_0 \lesssim r \ll r_{\rm tail}(E)$ where the kinetic energy is negligible with respect to the potential, namely $|E - E_{\rm o}| \ll |V_{\rm tail}(r_{\rm tail})|$. In that region, the open-channel wave functions are energy independent, i.e. all proportional to the threshold solution at $E = E_{\rm o}$. If the coupling occurs in that region, it is well known that one can employ the quantum defect theory (QDT) [13, 15–17, 27–34] to accurately describe the system for all the energies above and below the threshold $E_{\rm o}$ that are smaller than $|V_{\rm tail}(r_{\rm w})|$. Although the usual treatment of QDT makes use of the short-distance K and Y matrices, here all quantities shall be expressed in terms of observables such as $a_{\rm o}$ and $\gamma$. Doing so, one obtains a "renormalised" formulation of QDT.

Above the threshold, one finds that the scattering phase shift Eq. (2) is given by the following expressions for the shift and width (see Appendix D.1):

$$\Delta(k) = \Delta_0 + B_{\rm o}(k)\frac{\Gamma(k)}{2}\,, \qquad \frac{\Gamma(k)}{2} = \gamma\frac{k\,[A(k)]^{-2}}{1 + [B_{\rm o}(k)]^2}\,, \qquad (7)$$

with $B_{\rm o}(k) \equiv [\tan\bar\eta(k)]^{-1} - ka_{\rm o}\,[A(k)]^{-2}$, where $\bar\eta(k)$ and $A(k)$ are two dimensionless functions universally determined by the tail of $V_{\rm o}$ (see Appendix C.1). Physically, $\bar\eta$ is the difference $\eta^{(\infty)} - \eta^{(0)}$, where $\eta^{(a)}$ denotes the scattering phase shift for a potential with tail $V_{\rm tail}$ and scattering length $a$, and $A(k)$ is the amplitude of its radial wave function $u_\infty^{(k)}$ at infinite scattering length in the energy-independent region where $u_\infty^{(k)}(r) = A(k) \times u_\infty^{(0)}(r)$, with the zero-energy solution $u_\infty^{(0)}$ normalised so that $u_\infty^{(0)}(r) \xrightarrow[r\to\infty]{} 1$.

Below the threshold, one finds an even simpler result for the dressed bound state energy:

$$E_d = E_b + \Delta_0 + \frac{\gamma}{\lambda(\kappa) - a_o},$$

(8)

where the function $\lambda(\kappa)$ is determined purely from the tail of $V_o$ (see Appendix C.3). In fact, the energy $-\frac{\hbar^2 \kappa^2}{2\mu}$ as a function of $\lambda(\kappa)$ simply corresponds to the bound-state spectrum for a potential with tail $V_{\text{tail}}$ as a function of its scattering length. It is quite remarkable that the mere knowledge of the bare bound state spectrum for $V_o$ as a function of its scattering length entirely determines the dressed bound state spectrum through Eq. (8) once $\gamma$, $a_o$, and $\tilde{E}_b \equiv E_b + \Delta_0 - E_o$ are known.

Equations (7) and (8) constitute the first main result of this paper. They allow to determine the two-body observables for all energies above and below the threshold from only the three quantities $\gamma$, $a_o$, and $\tilde{E}_b$. Note that these quantities can be extracted from the zero-energy scattering length Eq. (3), within the approximation $a_{\text{bg}} \approx a_o$. It is therefore possible to determine a two-body observable (e.g. the dressed bound state energy) from the knowledge of another observable (e.g. the scattering length), without ever knowing the bare bound state causing the resonance, nor its coupling to the open channel.

The crucial point leading to this result is that the zero-energy shift $\Delta_0$ is taken apart and the width is expressed in terms of the reduced width $\gamma$. These are the only quantities that depend explicitly upon the three closed-channel parameters $W_0, a_c, a_c'$ through the expressions (see Appendix D.1),

$$\gamma = \frac{2\mu}{4\pi\hbar^2} W_0^2 (1 - a_o/a_c)^2, \qquad \Delta_0 = \gamma \frac{a_o - a_c - a_c'}{(a_o - a_c)^2}.$$

(9)

Here, $W_0$ characterises the strength of the coupling between the open channel and the bare bound state, while $a_c$ and $a_c'$ are two lengths characterising the closed channel. Like scattering lengths, both $a_c$ and $a_c'$ can be either positive or negative. The length $a_c$ was introduced in Ref. [21], which focused on a specific regime in which $a_c' = 0$ and $a_c$ sets the phase of oscillations of the bare bound state wave function in the coupling region. For this reason, $a_c$ was dubbed the "closed-channel scattering length", by analogy with the open-channel scattering length $a_o$ setting the phase of oscillations in the open channel. Note however that in general $a_c' \neq 0$ and $a_c$ cannot always be interpreted as a scattering length for the closed channel.

In this renormalised formulation where $\Delta_0$ and $\gamma$ are taken apart, one can now see the distinctive property of the QDT: since the closed-channel parameters $W_0, a_c, a_c'$ only affect the values of $\gamma$ and $\Delta_0$, they cannot be individually determined from the observables of Eq. (7) or (8). The resonance is thus largely independent of the details of the closed channel. This is in sharp contrast with Eqs. (5-6), which depend explicitly on the closed-channel parameter $\sigma$ even after taking apart the zero-energy width $\Delta_0$ and the reduced width $\gamma$.

## 4.2 Application to magnetic Feshbach resonances

The QDT typically applies to ultracold atoms undergoing a magnetic Feshbach resonance [3, 27, 33–35]. The QDT regime is reached due to the deep van der Waals tail $V_{\text{tail}}(r) = -C_6/r^6 = -\frac{\hbar^2}{2\mu}(2r_{\text{vdW}})^4/r^6$ of the interatomic interactions, where $r_{\text{vdW}}$ is the van der Waals length. In these systems, the bare bound energy $E_b$ in Eqs. (2,3,8) is related to the magnetic field intensity $B$ by the Zeeman shift through the relation $E_b + \Delta_0 - E_o = \delta\mu \times (B - B_0)$, where $\delta\mu$ is the magnetic moment difference between the open and closed channels, and $B_0$ is the magnetic field intensity at which the resonance is observed at the threshold. The reduced width is related to the observed magnetic width $\Delta B$ by $\gamma = a_{\text{bg}} \delta\mu \times \Delta B$. Thus, once the physical parameters $r_{\text{vdW}}$, $\delta\mu$, $\Delta B$, $B_0$, and $a_{\text{bg}} \approx a_o$ are known, all two-body observables can be determined from Eqs. (7) and (8).

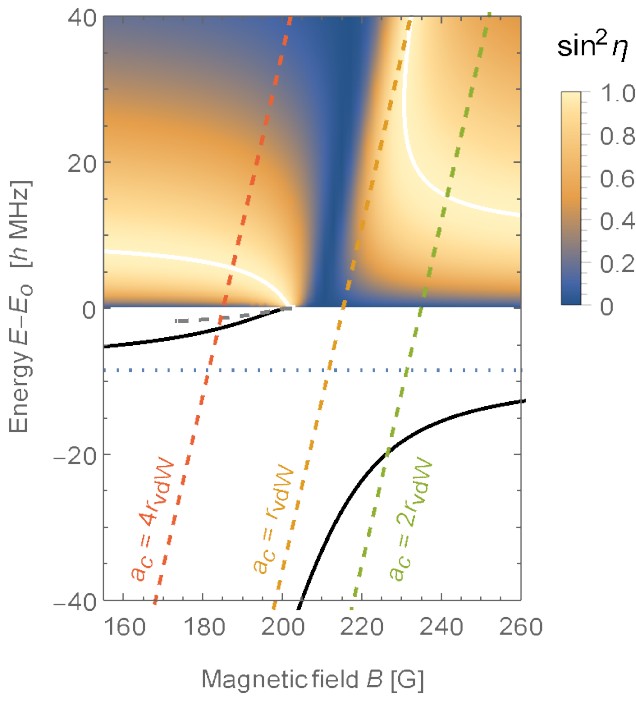

Figure 2: Energy spectrum and scattering phase shift of the $^{40}$K $ab$ Feshbach resonance near $B_0 = 202$ G. The solid curve below the continuum threshold shows the dressed bound state energy obtained from the QDT formula Eq. (8), and the shading above the continuum threshold shows the quantity $\sin^2 \eta$ obtained from Eq. (2) with the QDT formulas Eq. (7). The resonance ($\sin^2 \eta = 1$) is shown as a white curve. The functions $\lambda(\kappa)$, $\bar{\eta}(k)$, and $A(k)$ are obtained for a van der Waals tail $V_{\text{tail}}(r) = -\frac{\hbar^2}{2\mu}(2r_{\text{vdW}})^4/r^6$, and the following parameters are used: $\tilde{E}_b = E_b + \Delta_0 - E_o = \delta\mu \times (B - B_0)$ with $\delta\mu/h = 2.35$ MHz/G, $\gamma/h = 50$ MHz$\times r_{\text{vdW}}$, $a_{\text{bg}} = 2.635\, r_{\text{vdW}}$, and $E_{\text{vdW}}/h = \frac{\hbar}{4\pi\mu r_{\text{vdW}}^2} = 21$ MHz. The horizontal dotted line shows the bound state energy of the open channel. Note that the coupling of this bound state to the closed channel creates an avoided crossing that splits the dressed bound state energy curve into two branches: one on the left side which reaches the open-channel threshold at $B_0$, and one on the right side which asymptotes to the open-channel bound state energy. The dashed grey curve shows the dressed bound state energy obtained from the QDT Eq. (8) in the zero-range limit, corresponding to Eq. (11). This plot reproduces Fig. 13 of Ref. [3], except that the bare bound state energy $E_b$ is shifted. The dashed slanted lines show the bare bound state energy $E_b$ for three different models, whose closed-channel parameters are (arbitrarily) set to $a'_c = 0$ and $a_c = 4r_{\text{vdW}}, r_{\text{vdW}}, 2r_{\text{vdW}}$, respectively from left to right, and $W_0$ is set to maintain the same value of $\gamma$. All models reproduce exactly the same dressed energy and scattering phase shift. This shows that the position of the bare bound state is arbitrary and not constrained by the observables.

An example is shown in Fig. 2 for a resonance between $^{40}$K atoms. It is described by three models with different values of $W_0, a_c, a'_c$, but conforming to the same renormalised QDT given by Eqs. (7) and (8). Thus there is no way of determining the values of the closed-channel parameters from the observables shown in that figure. Of course, if one could alter $a_o$ independently of the other model parameters, then the values of $W_0$ and $a_c$ could be inferred from the change in $\gamma$ by virtue of Eq. (9) [21]. However this does not appear to be possible exper-

imentally, and in any case the value of $a'_c$ would remain undetermined. One must conclude that although the two-channel QDT provides an excellent description of isolated resonances, its closed-channel parameters $W_0, a_c, a'_c$ are not fully constrained by observables, and thus the shift $\Delta_0$ and the bare energy $E_b$ are ambiguous quantities.

Here, the conservative point of view is taken that only scattering phase shifts and bound state energies are fundamentally observable at the two-body level. Other short-distance quantities can be observed by involving a third body (such as a photon or another atom), as discussed in Sec. 6.

### 4.3 Application to low-energy resonances (zero-range limit)

The QDT also applies to any resonance whose energy is very close to the threshold. Indeed, for energies sufficiently close to the threshold, the wave functions are energy-independent within the range of interactions, because the potentials and couplings appear very deep compared with the considered energies. The QDT formalism can therefore be applied, and the energy-independent region can be approximated by a vanishingly small region compared to the typical extent of wave functions. In this limit, one obtains the analytic expressions (see Appendix C.4.3)

$$A(k) = 1, \qquad \bar{\eta}(k) = \pi/2, \tag{10}$$

$$\lambda(\kappa) = 1/\kappa. \tag{11}$$

This leads to a universal behaviour of near-threshold resonances that is independent of the closed channel's details.

This zero-range QDT regime is nothing but the oft-used "two-channel zero-range model" [36–39]. It is easy to check from Eq. (7) and (2) that the effective range in this regime always has a negative value — (see Appendix D.2),

$$r_{\text{eff}} = -2R_\star \left(1 - \frac{a_{\text{bg}}}{a}\right)^2, \tag{12}$$

where the length $R_\star = \hbar^2/(2\mu\gamma)$ characterises the width of the resonance [37]. This negative effective range corresponds to a limit commonly called "narrow" or "closed-channel dominated" Feshbach resonance [3] in the context of cold atoms, and is obtained when $R_\star$ is much larger than the range of interactions. Thus, if the resonance has in fact a positive effective range, the two-channel zero-range universal regime only applies at small energies where the effective range correction is negligible. For those small energies, it reduces to the single-channel zero-range universal regime that is parametrised by the scattering length only. This zero-range universality is well known both in ultracold-atom physics [40] and hadron physics [7, 9].

For instance, the zero-range universal limit can be seen in the case of the magnetic Feshbach resonance of Fig. 2: close to the threshold, the dressed bound state energy (solid black curve) approaches the universal limit (dashed grey curve) obtained from the QDT Eq. (8) with Eq. (11). The zero-range universal limit can also be seen in the Gaussian model of Fig. 1: as already mentioned in Sec. 3, close to the threshold, the curves for different values of $\sigma$ all coincide with the zero-range QDT (dashed grey curve) obtained with Eqs. (11). In this low-energy limit, the closed-channel parameter $\sigma$, and thus $\Delta_0$ and $E_b$, become irrelevant, just as in Sec. 4.2.

However, away from the zero-range universal regime, there is a clear discrepancy between magnetic Feshbach resonances and other kinds of Feshbach resonances.

On the one hand, magnetic Feshbach resonances remain described by the van der Waals QDT away from the threshold, since their interactions feature a deep van der Waals tail. This results in a dressed bound state (the single solid black curve of Fig. 2) that remains the same whatever the closed-channel parameters.

On the other hand, resonances with shallow interactions are not described by a QDT away from the threshold. Thus they become sensitive to the closed channel details, as illustrated by the several curves of Fig. 1 obtained for different closed-channel parameters. This is the case, for instance, for hadron resonances, since hadronic interactions feature a shallow tail [41]. Hence, the closed-channel parameters of hadron resonances that are not very close to the threshold could in principle be identified with enough data.

There have already been indications [26, 42–44] that some hadron resonances significantly deviate from the zero-range universal regime. For example, some resonances feature a positive effective range, which by construction cannot be reproduced by Eq. (12), or lead to an open-channel fraction $X$ (also called "compositeness" [45]) that exceeds unity when evaluated in the zero-range limit with $a_\text{o} = 0$. Table II in Ref. [44] lists several hadron resonances with their corresponding binding energy, scattering length, and effective range, obtained either from experimental data or ab initio calculations. These three quantities cannot in general be reproduced by the zero-range QDT with $a_\text{o} = 0$, because there are only two parameters in that theory, $\gamma$ and $\tilde{E}_\text{b}$.

One can of course include a non-zero scattering length $a_\text{o}$ in the open channel. For instance, in the case of the X(3872) state [46], suspected to result from a resonance between a pair of $D^0$ and $\bar{D}^{*0}$ mesons and a compact $\bar{c}c$ bare bound state [47], fitting the quantities $E_\text{d} - E_\text{o} = -18$ keV, $a = 28.5$ fm, and $r_\text{eff} = -5.34$ fm listed in Ref. [44], leads to $a_\text{o} = 25.3$ fm. Since this model is in a QDT regime, the fit does not provide any information about the bare bound state.

Eventually though, as more data is accumulated, it should prove impossible to reproduce all data with only the three parameters of the zero-range QDT, and models beyond it will become necessary. For example, one may fit the above data with the nonrelativistic Gaussian model of Eqs. (5-6). In this case, the extra parameter is given by the closed-channel parameter $\sigma$, and one finds $\sigma = 23.2$ fm. Since this model is not in a QDT regime, it allows to determine the mass of the bare bound state with respect to the $D^0$-$\bar{D}^{*0}$ threshold, namely $E_\text{b} - E_\text{o} = -10.1$ keV. Of course, the significance of this value is tied to one's trust in the model. The simplistic Gaussian model is unlikely to provide an adequate description of the X(3872) state, not to mention the complications related to the proximity of other thresholds and decay channels [47]. It nevertheless illustrates how a model beyond the zero-range QDT regime can extract some information about the compact core from experimental data.

## 5 Discussion

### 5.1 Closed-channel fraction

Let us now mention a remarkable point. While the properties of the bare bound state appear to be unobservable in the QDT regime, its proportion $Z = 1 - X$ in the dressed bound state (called the "closed-channel fraction" [19, 20, 48–52] in the context of ultracold-atom physics and "elementariness" [7, 26, 42, 43, 45, 53] in hadron physics) is observable and has indeed been measured in ultracold-atomic systems [48, 52]. It can be easily calculated from the Hellmann-Feynman theorem [51], yielding $Z = dE_\text{d}/dE_\text{b}$.

Quite naturally, the closed-channel fraction in general depends on the closed-channel details. For instance, for the Gaussian model, taking the derivative of Eq. (6) with respect to $E_d$ results in a closed-channel fraction $Z$ that explicitly depends on the parameter $\sigma$ and thus $\Delta_0$.

In contrast, in the QDT regime, taking the derivative of Eq. (8) with respect to $E_d$ gives an expression that is independent of the closed-channel parameters, and in particular of the shift $\Delta_0$. It may be surprising that the fraction of the bare bound state in the dressed wave function remains unaltered, even though the bare bound state energy itself can be arbitrarily shifted away by $\Delta_0$. For instance, one would intuitively think that the fraction goes to unity only when the dressed energy approaches the bare energy. However, the formula $Z = dE_d/dE_b$ shows that this is the case even when the two energy curves are parallel to each other. Physically, it means that even away from the resonance where the dressed bound state is almost purely in the bare state, its energy may be significantly shifted from the bare state energy through the coupling to the open channel. This reconciles the two facts that the closed-channel bare bound state is not directly observable but its fraction in the dressed bound state is.

Incidentally, one can also understand from these considerations that the intuitive picture according to which the dressed bound state results from an avoided crossing between the bare bound states in the open and closed channels does not always hold. For instance, in Fig. 2, the dressed bound state (solid black curve) appears to result from the avoided crossing between the open-channel bound state (dotted line) and the bound state in the closed channel (orange dashed line) corresponding to $a_c = r_{vdW}$. However, for the other values of $a_c$ leading to different bare bound state energies in the closed channel (dashed red or green curve), the avoided crossing picture is much less apparent, even though the observables remain the same. The reason is that an avoided crossing results from the coupling of only two discrete bound states, whereas here the continuum of states in the open channel can play a significant role and strongly alter the avoided crossing picture.

## 5.2 Dependence on the closed channel

Even though quantities such as $E_b$ and $\Delta_0$ are found to be ambiguous and non-observable in the QDT regime, they do have definite values for a given model, and these values depend on the closed-channel parameters. In particular, the following expression for the zero-energy shift $\Delta_0$ [3, 17, 20, 54–57],

$$\frac{\gamma}{\bar{a}} \frac{\frac{a_o}{\bar{a}} - 1}{1 + (\frac{a_o}{\bar{a}} - 1)^2} \,, \tag{13}$$

has been shown to be incorrect in Ref. [21], resulting from an invalid approximation in the QDT formalism. It can readily be seen that this expression depends only on $a_o$ and the characteristic range $\bar{a}$ of the open-channel potential $V_o(r)$, but has no dependence on the closed channel, in disagreement with the correct expression in Eq. (9). Nevertheless, since $\Delta_0$ is unobservable, it is always possible for fixed values of $\gamma$ and $a_o$ to devise a model with a choice of $W_0, a_c, a_c'$ satisfying Eq. (13), as done in Refs. [54, 57]. This arbitrary choice does not affect the two-body observables. Thus, while the value of $\Delta_0$ in Eq. (13) has no special significance, its use in these works has no consequence on two-body observables.

However, the works of Refs. [54, 57] are concerned with three-body systems. This raises the important question whether the value of $\Delta_0$, and more generally the closed-channel parameters, could affect three-body observables. Indeed, three-body observables are not only affected by two-body binding energies and scattering phase shifts, but also off-the-energy-shell two-body quantities, such as the short-distance two-body wave function. The next and final section investigates how short-distance observables are affected by the closed-channel parameters.

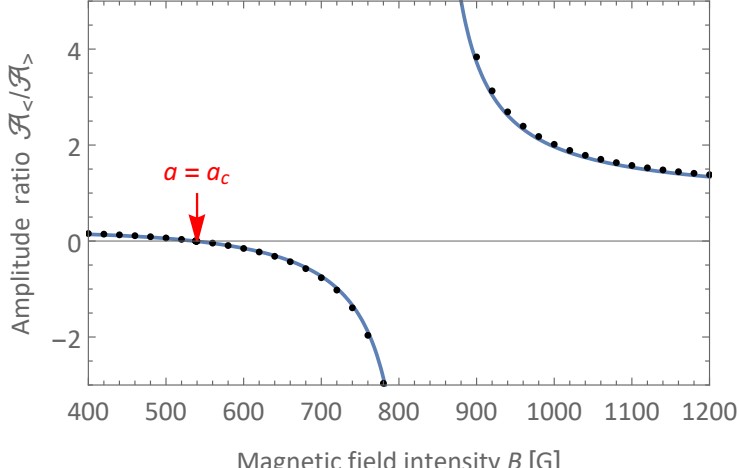

Figure 3: Ratio of the amplitudes $\mathcal{A}_<$ and $\mathcal{A}_>$ defined in Eq. (14) as a function of magnetic field, for the $^6$Li $ab$ resonance near $B_0 = 834$ G. The dots correspond to a realistic calculation with five hyperfine channels, where the two amplitudes are obtained from the triplet component of the zero-energy open-channel wave function. The solid curve represents Eq. (15) with the values $a_c = 2.391$ nm and $a_o = -112.8$ nm, corresponding respectively to the singlet and triplet scattering lengths. The scattering length $a$ is given by Eq. (3) with $a_{bg} = -84.89 - 24.19(B/B_0 - 1) + 22.77(B/B_0 - 1)^2$ nm, $\gamma/h = 62\,770$ MHz nm, and $\tilde{E}_b = E_b + \Delta_0 - E_o = \delta\mu \times (B - B_0)$ with $\delta\mu/h = 2.8$ MHz/G.

# 6 Short-distance physics

The QDT gives a simple account of the short-distance two-body physics. The wave function can be expressed by its open-channel radial component $u_o(r)$ and closed-channel radial component $u_c(r)$. In the energy independent region $r_0 \lesssim r \ll r_{tail}(E)$, one finds that $u_o(r)$ exhibits oscillations with a different phase and amplitude beneath and beyond the coupling distance $r_w$ (see Appendix D.3):

$$u_o(r) = \begin{cases} \mathcal{A}_< \times u_{a_o}^{(0)}(r), & r \ll r_w, \\ \mathcal{A}_> \times u_{a_{eff}}^{(0)}(r), & r \gg r_w, \end{cases} \tag{14}$$

where $u_a^{(0)}(r) \xrightarrow[r\to\infty]{} r - a$ is the zero-energy solution of the open-channel potential with scattering length $a$, and $a_{eff}$ is the energy-dependent scattering length $a_{bg} + \gamma/(E - E_b - \Delta_0)$. This shows that the open-channel wave function has an unperturbed form with amplitude $\mathcal{A}_<$ beneath the coupling region, and a perturbed form with amplitude $\mathcal{A}_>$ beyond the coupling region.

At low energy, the ratio of the amplitudes $\mathcal{A}_</\mathcal{A}_>$ is given by:

$$\boxed{\frac{\mathcal{A}_<}{\mathcal{A}_>} = \frac{a - a_c}{a_o - a_c}.} \tag{15}$$

This formula constitutes the second main result of this paper. It gives a physical interpretation of $a_c$ as the scattering length at which the short-distance amplitude $\mathcal{A}_<$ vanishes.

This dependence of the short-range amplitudes on the closed-channel scattering length $a_c$ is illustrated in Fig. 3 for the case of the 834 G magnetic Feshbach resonance between $^6$Li atoms. This case is fortunate, since a two-channel model has been clearly identified for this resonance

as originating from the coupling of a spin triplet open channel with a spin singlet bare bound state. This suggests a photoassociation experiment to measure $a_c$: by photoassociating $^6$Li atoms to an excited triplet bound state with an extent smaller than $r_w$, one could measure $\mathcal{A}_<$ from the photoassociation signal, and determine $a_c$ from the magnetic field at which $\mathcal{A}_<$ vanishes. For other multi-channel resonances, however, it remains a challenge to identify the effective two-channel model in general.

## 7 Conclusion

In summary, this work clarifies the role of closed-channel parameters in Feshbach resonances. On the one hand, it is found that they affect two-body observables (scattering phase shifts and binding energies) in the general case, but not in the case of resonances involving deep interaction potentials, such as magnetic Feshbach resonances between ultracold atoms. Thus, the closed-channel parameters of magnetic Feshbach resonances cannot be determined from these observables. This is in sharp contrast with resonances involving shallow interaction potentials, such as hadron resonances, for which this situation occurs only close to the open-channel threshold.

On the other hand, one of the closed-channel parameters, called the "closed-channel scattering length", is found to affect short-distance two-body physics. In ultracold-atomic systems, this parameter could be determined by photoassociation, and should also affect three-body observables, such as three-body recombination loss rates. The closed-channel scattering length could thus play a role in the determination of the three-body parameter characterising the Efimov spectrum of three-body states near a magnetic Feshbach resonance [58], which has been measured in various experiments, and for which a full theoretical understanding is still in progress [57,59–62].

## Acknowledgments

The author is grateful to P. S. Julienne, E. Tiesinga, L. Pricoupenko, M. Raoult, S. Kokkelmans, N. Kjærgaard, M. Oka, S. Endo, T. Hyodo, and T. Kinugawa for stimulating discussions on this topic. The author is especially thankful to S. Endo for carefully checking the formulas in this paper.

**Funding information** This work was supported by the JSPS Kakenhi grant No. JP23K03292.

## A Two-channel model

The Hamiltonian for a two-channel model of a two-particle system reads,

$$H = \left( \begin{array}{cc} H_{oo} & H_{oc} \\ H_{co} & H_{cc} \end{array} \right), \tag{A.1}$$

where the open-channel Hamiltonian $H_{oo}$ and the closed-channel Hamiltonian $H_{cc}$ are given by

$$H_{oo} = T + V_o, \tag{A.2}$$

$$H_{cc} = T + V_c, \tag{A.3}$$

where $T$ is the relative kinetic operator, which for non-relativistic systems is given by $\langle \boldsymbol{p}|T|\boldsymbol{q}\rangle = \frac{\hbar^2 p^2}{2\mu}\delta^3(\boldsymbol{p}-\boldsymbol{q})$ where $\mu$ is the reduced mass of the particles. The open-channel interaction potential $V_{\mathrm{o}}$ asymptotes to a certain energy threshold $E_{\mathrm{o}}$ with a potential tail $V_{\mathrm{tail}}$, i.e. $V_{\mathrm{o}}(r) \xrightarrow[r\to\infty]{} E_{\mathrm{o}} + V_{\mathrm{tail}}(r)$. The closed-channel potential $V_{\mathrm{c}}$ asymptotes to a certain energy $E_{\mathrm{c}} > E_{\mathrm{o}}$.

The wave function $\phi$ of the system has two components, $\phi_{\mathrm{o}}$ and $\phi_{\mathrm{c}}$, respectively for the open and closed channels. At energy $E$, they satisfy the coupled Schrödinger equations,

$$(T + V_{\mathrm{o}} - E)|\phi_{\mathrm{o}}\rangle + H_{\mathrm{oc}}|\phi_{\mathrm{c}}\rangle = 0\,, \tag{A.4}$$

$$(T + V_{\mathrm{c}} - E)|\phi_{\mathrm{c}}\rangle + H_{\mathrm{co}}|\phi_{\mathrm{o}}\rangle = 0\,. \tag{A.5}$$

For energy $E < E_{\mathrm{c}}$ (such that the second channel is indeed closed), these equations lead to:

$$|\phi_{\mathrm{o}}\rangle = |\bar{\phi}_{\mathrm{o}}^{E,\hat{k}}\rangle + G_{\mathrm{o}}^+ H_{\mathrm{oc}}|\phi_{\mathrm{c}}\rangle\,, \tag{A.6}$$

$$|\phi_{\mathrm{c}}\rangle = G_{\mathrm{c}} H_{\mathrm{co}}|\phi_{\mathrm{o}}\rangle\,, \tag{A.7}$$

where $G_{\mathrm{o}}^+ = (E + i0^+ - T - V_{\mathrm{o}})^{-1}$ and $G_{\mathrm{c}} = (E - T - V_{\mathrm{c}})^{-1}$ are the resolvents of the open and closed channels, and $|\bar{\phi}_{\mathrm{o}}^{E,\hat{k}}\rangle$ is the scattering eigenstate of the open-channel Hamiltonian at energy $E$ and scattering direction $\hat{k}$, normalised as $\langle \bar{\phi}_{\mathrm{o}}^{E,\hat{k}}|\bar{\phi}_{\mathrm{o}}^{E',\hat{k}'}\rangle = \delta(E - E')\delta(\hat{k} - \hat{k}')$.

## B  Two-channel isolated resonance theory

### B.1  Definition of the resonance shift and width

The closed-channel potential $V_{\mathrm{c}}$ is assumed to support a bound state $|\phi_{\mathrm{b}}\rangle$ with energy $E_{\mathrm{b}}$:

$$H_{\mathrm{cc}}|\phi_{\mathrm{b}}\rangle = E_{\mathrm{b}}|\phi_{\mathrm{b}}\rangle\,. \tag{B.1}$$

It is normalised as $\langle \phi_{\mathrm{b}}|\phi_{\mathrm{b}}\rangle = 1$. In the isolated resonance approximation, only this bound state gives a significant contribution to the resonance, so that one may write:

$$G_{\mathrm{c}} = \frac{|\phi_{\mathrm{b}}\rangle\langle\phi_{\mathrm{b}}|}{E - E_{\mathrm{b}}} + G_{\mathrm{c}}^{\mathrm{nr}}\,, \tag{B.2}$$

where the non-resonant part $G_{\mathrm{c}}^{\mathrm{nr}}$ only gives a small contribution from the other states of the closed channel. This leads to:

$$|\phi_{\mathrm{o}}\rangle = |\phi_{\mathrm{bg}}\rangle + \frac{G_{\mathrm{o}}^+ |W\rangle\langle W|\phi_{\mathrm{bg}}\rangle}{E - E_{\mathrm{b}} - \Delta^+}\,, \tag{B.3}$$

$$|\phi_{\mathrm{c}}\rangle = |\phi_{\mathrm{b}}\rangle \frac{\langle W|\phi_{\mathrm{bg}}\rangle}{E - E_{\mathrm{b}} - \Delta^+} + G_{\mathrm{c}}^{\mathrm{nr}} H_{\mathrm{co}}|\phi_{\mathrm{o}}\rangle\,, \tag{B.4}$$

with the short-hand notations

$$|W\rangle \equiv H_{\mathrm{oc}}|\phi_{\mathrm{b}}\rangle\,, \tag{B.5}$$

$$|\phi_{\mathrm{bg}}\rangle \equiv |\bar{\phi}_{\mathrm{o}}^{E,\hat{k}}\rangle + G_{\mathrm{o}}^+ H_{\mathrm{oc}} G_{\mathrm{c}}^{\mathrm{nr}} H_{\mathrm{co}}|\phi_{\mathrm{o}}\rangle\,, \tag{B.6}$$

and

$$\boxed{\Delta^+ \equiv \langle W|G_{\mathrm{o}}^+|W\rangle \equiv \Delta - i\frac{\Gamma}{2}\,,} \tag{B.7}$$

which defines the energy-dependent shift $\Delta(E)$ and width $\Gamma(E)$.

## B.2 Partial wave expansion

Combining Eqs. (A.6-A.7) gives a closed equation on $\phi_o$:

$$|\phi_o\rangle = |\bar{\phi}_o^{E,\hat{k}}\rangle + G_o^+ H_{oc} G_c H_{co} |\phi_o\rangle.$$

Making the partial wave expansion along the direction $\hat{k}$ of the incoming wave,

$$\langle \boldsymbol{r}|\phi_o\rangle \equiv \sum_\ell \frac{\phi_{o,\ell}(r)}{r} Y_{\ell 0}(\hat{r}), \tag{B.8}$$

$$\langle \boldsymbol{r}|\bar{\phi}_o^{E,\hat{k}}\rangle \equiv \sum_\ell \frac{\bar{\phi}_{o,\ell}(r)}{r} Y_{\ell 0}(\hat{r}), \tag{B.9}$$

$$\langle \boldsymbol{r}|H_{oc}G_c H_{co}|\boldsymbol{r}'\rangle \equiv \sum_\ell \frac{H_\ell(r,r')}{rr'} Y_{\ell 0}(\hat{r}) Y_{\ell 0}^*(\hat{r}'), \tag{B.10}$$

one finds for each partial wave $\ell$ the following complex radial wave equation:

$$\phi_{o,\ell}(r) = \bar{\phi}_{o,\ell}(r) + \int_0^\infty dr' g_{o,\ell}^+(r,r') \int_0^\infty dr'' H_\ell(r',r'') \phi_{o,\ell}(r''), \tag{B.11}$$

where the retarded partial-wave Green's function $g_{o,\ell}^+$ is given by

$$g_{o,\ell}^+(r,r') = -\frac{2\mu}{\hbar^2 k} \bar{u}_o(r_<) \bar{v}_o^+(r_>), \tag{B.12}$$

with $k = \sqrt{2\mu(E-E_o)}/\hbar$, $r_> = \max(r,r')$ and $r_< = \min(r,r')$. The two functions $\bar{u}_o$ and $\bar{v}_o^+ \equiv \bar{v}_o + i\,\bar{u}_o$ are two independent solutions of the partial-wave radial equation:

$$\left(-\frac{d^2}{dr^2} + \frac{\ell(\ell+1)}{r^2} + \frac{2\mu}{\hbar^2}[V_o(r) - E_o] - k^2\right) u(r) = 0, \tag{B.13}$$

satisfying

$$\bar{u}_o(r) \xrightarrow[r\to\infty]{} \sin(kr - \ell\pi/2 + \eta_o), \tag{B.14}$$

$$\bar{v}_o(r) \xrightarrow[r\to\infty]{} \cos(kr - \ell\pi/2 + \eta_o), \tag{B.15}$$

where $\eta_o$ is the $\ell$-wave scattering phase shift of the open channel. The solution $\bar{u}_o(r)$ is regular (vanishing when $r \to 0$), whereas the solutions $\bar{v}_o^+(r)$ and $\bar{v}_o(r)$ are irregular (non-vanishing for $r \to 0$).

In the following, the notations $(A|B) \equiv \int dr A(r) B(r)$ and $(A|B|C) \equiv \int dr \int dr' A(r) B(r,r') C(r')$ will used.

From the definitions of $\bar{\phi}_o^{E,\hat{k}}$ and $\bar{u}_o$, one finds

$$\bar{\phi}_{o,\ell}(r) = \bar{\mathcal{N}}_\ell \times \bar{u}_o(r), \tag{B.16}$$

with the complex coefficient $\bar{\mathcal{N}}_\ell \equiv \sqrt{\frac{2\mu(2\ell+1)}{\pi\hbar^2 k}} i^\ell e^{i\eta_o}$. The complex equation Eq. (B.11) can then be made real by splitting the real and imaginary parts of the Green's function Eq. (B.12), and setting

$$\phi_{o,\ell}(r) \equiv \mathcal{N}_\ell \times u_o(r), \tag{B.17}$$

with the complex coefficient $\mathcal{N}_\ell \equiv \bar{\mathcal{N}}_\ell \left(1 + i\frac{2\mu}{\hbar^2 k}(\bar{u}_{\mathrm{o}}|H_\ell|u_{\mathrm{o}})\right)^{-1}$. This yields the following equation for the real radial wave function $u_{\mathrm{o}}$:

$$u_{\mathrm{o}}(r) = \bar{u}_{\mathrm{o}}(r) + \int_0^\infty dr' g_{\mathrm{o},\ell}(r,r') \int_0^\infty dr'' H_\ell(r',r'') u_{\mathrm{o}}(r''), \tag{B.18}$$

with the non-retarded partial-wave Green's function,

$$g_{\mathrm{o},\ell}(r,r') \equiv -\frac{2\mu}{\hbar^2 k}\, \bar{u}_{\mathrm{o}}(r_<)\, \bar{v}_{\mathrm{o}}(r_>). \tag{B.19}$$

### B.3 Isolated resonance

#### B.3.1 Scattering phase shift

Using the isolated resonance decomposition Eq. (B.2) in Eq. (B.18), and assuming that $|W\rangle = H_{\mathrm{oc}}|\phi_{\mathrm{b}}\rangle$ is of the form

$$\langle \boldsymbol{r}|W\rangle = \frac{w(r)}{r} Y_{\ell 0}(\hat{r}), \tag{B.20}$$

acting on a specific partial wave $\ell$, one obtains for that partial wave:

$$H_\ell(r,r') = \frac{w(r)w(r')}{E - E_b} + H_\ell^{\mathrm{nr}}(r,r'), \tag{B.21}$$

where $H_\ell^{\mathrm{nr}}$ correspond to the non-resonant part $H_{\mathrm{oc}}G_{\mathrm{c}}^{\mathrm{nr}}H_{\mathrm{co}}$. This gives

$$u_{\mathrm{o}}(r) = u_{\mathrm{bg}}(r) + (w|u_{\mathrm{o}})\frac{\int_0^\infty dr' g_{\mathrm{o},\ell}(r,r')w(r')}{E - E_b}, \tag{B.22}$$

with the background function

$$u_{\mathrm{bg}}(r) \equiv \bar{u}_{\mathrm{o}}(r) + \int_0^\infty dr' g_{\mathrm{o},\ell}(r,r')w_{\mathrm{nr}}(r'), \tag{B.23}$$

where

$$w_{\mathrm{nr}}(r) = \int_0^\infty dr' H_\ell^{\mathrm{nr}}(r,r') u_{\mathrm{o}}(r'), \tag{B.24}$$

corresponds to the coupling to other states than the bare bound state causing the resonance.

Applying $(w|$ to the left of Eq. (B.22) to find $(w|u_{\mathrm{o}})$, and inserting the result back into Eq. (B.22) gives

$$u_{\mathrm{o}}(r) = u_{\mathrm{bg}}(r) + (w|u_{\mathrm{bg}})\frac{\int_0^\infty dr' g_{\mathrm{o},\ell}(r,r')w(r')}{E - E_{\mathrm{b}} - \Delta}, \tag{B.25}$$

with the shift $\Delta = (w|g_{\mathrm{o},\ell}|w)$. At large distances, the radial wave function becomes

$$u_{\mathrm{o}}(r) = \bar{u}_{\mathrm{o}}(r) - \left[\xi_{\mathrm{nr}} + \frac{(\Gamma + \Gamma_{\mathrm{nr}})/2}{E - E_{\mathrm{b}} - \Delta}\right]\bar{v}_{\mathrm{o}}(r), \tag{B.26}$$

with the width $\Gamma/2 = \frac{2\mu}{\hbar^2 k}|(w|\bar{u}_{\mathrm{o}})|^2$ and the non-resonant corrections

$$\Gamma_{\mathrm{nr}}/2 \equiv \frac{2\mu}{\hbar^2 k}(w|g_{\mathrm{o},\ell}|w_{\mathrm{nr}})(\bar{u}_{\mathrm{o}}|w), \tag{B.27}$$

$$\xi_{\mathrm{nr}} \equiv \frac{2\mu}{\hbar^2 k}(\bar{u}_{\mathrm{o}}|w_{\mathrm{nr}}). \tag{B.28}$$

Using the asymptotic behaviours of $\bar{u}_{\rm o}$ and $\bar{v}_{\rm o}$ given in Eqs. (B.14-B.15), one obtains from Eq. (B.26),

$$u_{\rm o}(r) \xrightarrow[r \to \infty]{} \propto \sin(kr - \ell\pi/2 + \eta),$$  (B.29)

with the scattering phase shift,

$$\eta = \eta_{\rm o} - \arctan\left(\xi_{\rm nr} + \frac{(\Gamma + \Gamma_{\rm nr})/2}{E - E_{\rm b} - \Delta}\right).$$  (B.30)

Treating the non-resonant corrections as a first-order perturbation, one finally arrives at

$$\boxed{\eta = \eta_{\rm bg} - \arctan\frac{\tilde{\Gamma}/2}{E - E_{\rm b} - \Delta}},$$  (B.31)

with the background phase shift:

$$\eta_{\rm bg} \equiv \eta_{\rm o} - \xi_{\rm nr}\left[1 + \left(\frac{\Gamma/2}{E - E_{\rm b} - \Delta}\right)^2\right]^{-1},$$  (B.32)

and the corrected width:

$$\tilde{\Gamma} \equiv \Gamma + \Gamma_{\rm nr}.$$  (B.33)

In the fully isolated resonance approximation, one neglects the non-resonant corrections, yielding $\eta_{\rm bg} \approx \eta_{\rm o}$ and $\tilde{\Gamma} \approx \Gamma$ in Eq. (B.31).

### B.3.2 Low-energy limit in the s wave

In the case of s wave ($\ell = 0$), the quantities $\Gamma, \Gamma_{\rm nr}$, and $\xi_{\rm nr}$ for small $k$ are proportional to $k$ (being proportional to $\bar{u}_{\rm o}$) and thus one obtains the s-wave scattering length:

$$\boxed{a = -\lim_{k \to 0} \eta/k = a_{\rm bg} - \frac{\tilde{\gamma}}{E_b + \Delta_0 - E_{\rm o}}},$$  (B.34)

where $a_{\rm bg} = -\lim_{k \to 0} \eta_{\rm bg}/k = a_{\rm o} + a_{\rm nr}$ with $a_{\rm nr} = \lim_{k \to 0} \xi_{\rm nr}/k$, and $\tilde{\gamma} = \lim_{k \to 0} \tilde{\Gamma}/2k = \gamma + \gamma_{\rm nr}$, with $\gamma_{\rm nr} \equiv \lim_{k \to 0} \Gamma_{\rm nr}/2k$.

One can more generally assume that:

$$\Gamma_k/2 = \gamma k\left(1 + \beta k^2\right) + O(k^3),$$  (B.35)

$$\tilde{\Gamma}_k/2 = \tilde{\gamma} k\left(1 + \tilde{\beta} k^2\right) + O(k^3),$$  (B.36)

$$\Delta_k = \Delta_0 + \alpha k^2 + O(k^3),$$  (B.37)

$$\frac{k}{\tan\eta_{\rm bg}} = -\frac{1}{a_{\rm bg}} + \frac{1}{2}r_{\rm bg}k^2 + O(k^3),$$  (B.38)

so that one finds from Eq. (B.31) the following low-energy expansion:

$$\frac{k}{\tan\eta} = -\frac{1}{a} + \frac{1}{2}r_{\rm eff}k^2 + O(k^3),$$  (B.39)

with $a$ given by Eq. (B.34) and the effective range $r_{\rm eff}$ given by:

$$\boxed{r_{\rm eff} = 2\left(\frac{\alpha}{\tilde{\gamma}} - R_\star + \frac{a a_{\rm bg} + \tilde{\beta}}{a - a_{\rm bg}}\right)\left(1 - \frac{a_{\rm bg}}{a}\right)^2 + r_{\rm bg}\frac{a_{\rm bg}^2}{a^2}},$$  (B.40)

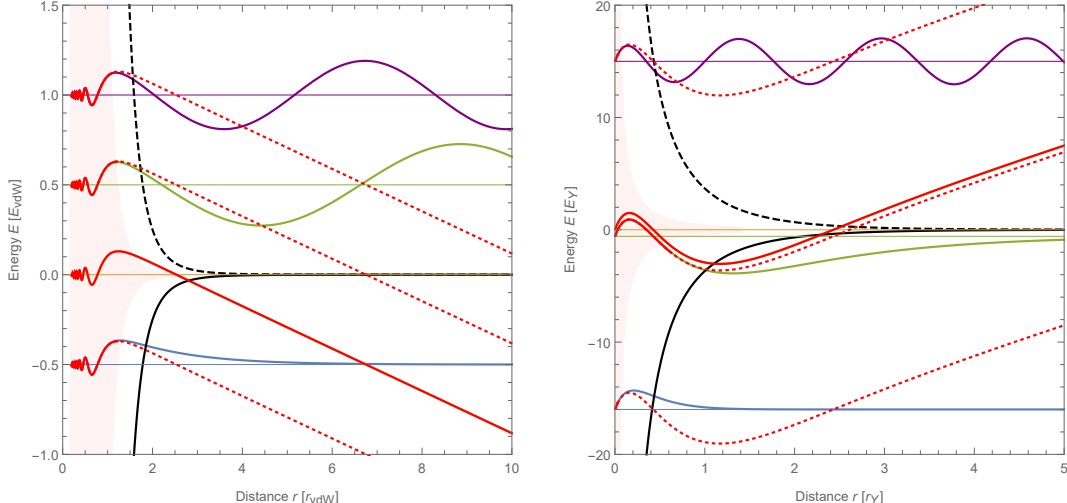

**Figure 4:** Energy independence of s-wave radial wave functions for two potentials: a deep van der Waals potential $V(r) = -E_{\text{vdW}} \left( \frac{r_{\text{vdW}}}{r} \right)^6$ with many bound states (left panel) and a shallow Yukawa potential $V(r) = -10 E_{\text{Y}} \frac{\exp(-r/r_{\text{Y}})}{r/r_{\text{Y}}}$ with only two bound states (right panel). Here, $r_{\text{vdW}}$ and $r_{\text{Y}}$ are the van der Waals and Yukawa ranges, and $E_{\text{vdW}} \equiv \frac{\hbar^2}{2\mu r_{\text{vdW}}^2}$ and $E_{\text{Y}} \equiv \frac{\hbar^2}{2\mu r_{\text{Y}}^2}$ are their associated energies. Each panel shows $V(r)$ (black curve) and $-V(r)$ (black dashed curve), along with the radial wave functions at selected energies (the curves are shifted according to their respective energies). The zero-energy radial wave function is shown in solid red, and superimposed as a red dotted curve onto the curves corresponding to radial wave functions at other energies. The region of energy independence $r_0 \lesssim r \ll r_{\text{tail}}$ (or equivalently $r \gtrsim r_0$ and $E \ll |V(r)|$) is shown as a pink shaded area. In that region, the zero-energy wave function matches the wave functions at other energies, and is shown in solid red. The energy independent region is much smaller in the case of a shallow potential, for which the quantum defect theory is barely applicable, except at very small energies $\ll E_{\text{Y}}$ and large distances $\gg r_{\text{Y}}$ corresponding to the zero-range limit. For instance, the last bound state is well described by the zero-range limit for $r \gg r_{\text{Y}}$, because it can be determined from a boundary condition at $r \approx r_{\text{Y}}$ that is similar to that of the zero-energy state. That is not the case for the lowest bound state, which differs too much from the zero-energy state.

where the following length [37] is introduced:

$$R_\star \equiv \frac{\hbar^2}{2\mu\tilde{\gamma}} \,. \tag{B.41}$$

Close to the resonance ($a \to \infty$), the effective range reduces to:

$$r_{\text{eff}} = 2\left( \frac{\alpha}{\tilde{\gamma}} + a_{\text{bg}} - R_\star \right) \,. \tag{B.42}$$

## C Quantum defect theory

The key point of the quantum defect theory is that when an interaction potential is sufficiently deep in a certain region $r_0 \lesssim r \ll r_{\text{tail}}$, the wave functions in that region are nearly

energy-independent for a range of energies that remain much smaller than the potential energy $V(r_\text{tail})$. In that region (and only in that region), the wave function at any of these energies is accurately described by a superposition of two independent solutions of the potential at zero energy. For a specific choice of two reference solutions, there is a particular linear combination reproducing the wave function in the region. The coefficients of this linear combination can be parametrised by a global normalisation factor, and a parameter called the *quantum defect*.

In the following, the reference functions are chosen as the two s-wave radial solutions of the potential at zero-energy, with respectively zero and infinite scattering length:

$$f_0(r) \xrightarrow[r\to\infty]{} r\,, \tag{C.1}$$

$$f_\infty(r) \xrightarrow[r\to\infty]{} 1\,. \tag{C.2}$$

The zero-energy solution with scattering length $a$ is thus $f_0(r) - a f_\infty(r) \xrightarrow[r\to\infty]{} r - a$. With this choice, the quantum defect is simply the s-wave scattering length $a$.

For example, for a van der Waals interaction $V(r) \to -C_6/r^6$, one has:

$$f_0(r) = r_\text{vdW}\sqrt{x}\,\Gamma(3/4)J_{-1/4}(2x^{-2})\,, \tag{C.3}$$

$$f_\infty(r) = \sqrt{x}\,\Gamma(5/4)J_{1/4}(2x^{-2})\,, \tag{C.4}$$

where $x = r/r_\text{vdW}$ and $r_\text{vdW}$ is the van der Waals length $r_\text{vdW} \equiv \frac{1}{2}\left(2\mu C_6/\hbar^2\right)^{1/4}$. The range of energy-independence at energy $|E| = \hbar^2 k^2/2\mu$ is given by $r_0 \lesssim r \ll r_\text{tail}$ with $r_\text{tail} = r_\text{vdW}^{2/3} k^{-1/3}$. It is illustrated in the left panel of Fig. 4 as a pink shaded area.

Interestingly, the quantum defect approach also applies to contact interactions. In this case, the region of energy-independence is restricted to the neighbourhood of $r = 0$ (i.e. $r_0 = r_\text{tail} = 0$) but extends to any energy. The two reference solutions are simply $f_0(r) = r$ and $f_\infty(r) = 1$. This is of course an idealisation, which can be regarded as the limit of a short-range interaction potential with vanishing range and infinite depth. Physically, it describes the wave functions of a short-range interaction potential for energies much smaller than the potential depth and distances larger than the potential range. The energy independent region in this case corresponds to energies smaller than the potential depth and distances smaller than the potential range. This is illustrated in the right panel of Fig. 4 for a shallow Yukawa potential. In the contact interaction limit, this region reduces to a boundary condition on the logarithmic derivative at $r = 0$.

## C.1 Positive energy

Let us now consider a potential $V(r) \xrightarrow[r\to\infty]{} 0$ of s-wave scattering length $a$ and its regular and irregular radial solutions $\bar{u}_a$ and $\bar{v}_a$ in the $\ell$th partial wave at finite positive energy $E = \hbar^2 k^2/2\mu > 0$. The regular function $\bar{u}_a$ is defined such that $\bar{u}_a(0) = 0$, which gives at large distance $\bar{u}_a(r) \to \sin(kr + \eta_a - \ell\pi/2)$ where $\eta_a$ is the scattering phase shift. The irregular solution $\bar{v}_a$ is chosen such that its phase at large distances is shifted by $\pi/2$ with respect to $\bar{u}_a$.

$$\bar{u}_a(r) \xrightarrow[r\to\infty]{} \sin(kr + \eta_a - \ell\pi/2)\,, \tag{C.5}$$

$$\bar{v}_a(r) \xrightarrow[r\to\infty]{} \cos(kr + \eta_a - \ell\pi/2)\,. \tag{C.6}$$

According to the quantum defect assumption, in the energy independent region $r_0 \lesssim r \ll r_\text{tail}$ the two functions $\bar{u}_a$ and $\bar{v}_a$ are linear combinations of the two zero-energy reference solutions $f_0$ and $f_\infty$. The regular solution $\bar{u}_a$ is simply proportional to the zero-energy solution $f_0 - a f_\infty$ with scattering length $a$:

$$\bar{u}_a(r) \xrightarrow[r_0\lesssim r\ll r_\text{tail}]{} D_a(k)(f_0(r) - a f_\infty(r))\,. \tag{C.7}$$

Similarly, the irregular solution $\bar{v}_a$ has the form:

$$\bar{v}_a(r) \xrightarrow[r_0 \lesssim r \ll r_{\text{tail}}]{} P_a(k)(f_0(r) - b_a(k)f_\infty(r)) . \tag{C.8}$$

The Wronskian $W[\bar{u}_a, \bar{v}_a] = \bar{u}_a(\bar{v}_a)' - (\bar{u}_a)'\bar{v}_a$ has the conserved value $-k$ calculated from Eqs. (C.5-C.6), so from the expressions of Eqs. (C.7-C.8), one finds:

$$\boxed{[b_a(k) - a]D_a(k)P_a(k) = -k,} \tag{C.9}$$

which shows that only two of the functions $D_a, b_a, P_a$ are independent for a given $a$.

One can determine $\eta_a, D_a, b_a$, and $P_a$ for any scattering length $a$, by just knowing four functions of $k$: $\eta_0, \eta_\infty, D_0, A$.

$$\tan \eta_a = \frac{(D_0)^{-1} \sin \eta_0 - a (A)^{-1} \sin \eta_\infty}{(D_0)^{-1} \cos \eta_0 - a (A)^{-1} \cos \eta_\infty}, \tag{C.10}$$

$$D_a = \left[ (D_0)^{-2} + \left(\frac{a}{A}\right)^2 - 2a \frac{\cos \bar{\eta}}{D_0 A} \right]^{-1/2}, \tag{C.11}$$

$$b_a = \frac{(A/D_0) - a \cos \bar{\eta}}{\cos \bar{\eta} - a (D_0/A)}, \tag{C.12}$$

$$P_a = -\frac{\cos \bar{\eta} - a (D_0/A)}{\sin \bar{\eta}} D_a, \tag{C.13}$$

with the notations

$$\bar{\eta} \equiv \eta_\infty - \eta_0, \tag{C.14}$$

and

$$\eta_0 \equiv \lim_{a \to 0} \eta_a, \qquad D_0 \equiv \lim_{a \to 0} D_a, \tag{C.15}$$

$$\eta_\infty \equiv \lim_{a \to -\infty} \eta_a, \qquad A \equiv \lim_{a \to -\infty} -aD_a. \tag{C.16}$$

Again the four functions $\eta_0, \eta_\infty, D_0, A$ are not independent, because the Wronskian $W[\bar{u}_0, \bar{u}_\infty]$ can be expressed at short distance as

$$W[D_0 f_0, A f_\infty] = D_0 A \underbrace{W[f_0, f_\infty]}_{-1} = -D_0 A, \tag{C.17}$$

and at large distance as:

$$W[\sin(kr - \ell\pi/2 + \eta_0), \sin(kr - \ell\pi/2 + \eta_\infty)] = -k \sin(\eta_\infty - \eta_0), \tag{C.18}$$

leading to the relation,

$$\boxed{D_0(k)A(k) = k \sin \bar{\eta}(k).} \tag{C.19}$$

Using this relation, one can express $D_a, b_a, P_a$ in terms of only two functions $A$ and $\bar{\eta}$:

$$D_a = \frac{k}{A} \left[ 1 + (B_a)^2 \right]^{-1/2}, \tag{C.20}$$

$$b_a = \frac{(A)^2}{k} \left[ \frac{1}{B_a} + \frac{1}{\tan \bar{\eta}} \right], \tag{C.21}$$

$$P_a = -B_a D_a, \tag{C.22}$$

where

$$B_a \equiv \frac{1}{\tan \bar{\eta}} - \frac{ka}{(A)^2}. \tag{C.23}$$

## C.2 Alternative choice

One may consider an alternative choice $\bar{v}_a^+(r) \equiv \bar{v}_a(r) + i\bar{u}_a(r)$ for the irregular function, that has the complex asymptote:

$$\bar{v}_a^+(r) \xrightarrow[r\to\infty]{} e^{i(kr+\eta_a-\ell\pi/2)}. \tag{C.24}$$

It can be expanded on $f_0$ and $f_\infty$

$$\bar{v}_a^+(r) \xrightarrow[r_0\lesssim r\ll r_{\text{tail}}]{} P_a^+ \left( f_0(r) - b_a^+ f_\infty(r) \right), \tag{C.25}$$

where the complex quantities $P_a^+$ and $b_a^+$ are readily obtained from Eqs. (C.7-C.8):

$$P_a^+ = P_a + iD_a, \tag{C.26}$$

$$b_a^+ = \frac{P_a b_a + iD_a a}{P_a + iD_a}. \tag{C.27}$$

The interest of this alternative choice is that the quantity $b_a^+$ is independent of $a$. Indeed, using Eqs. (C.11-C.13), one finds:

$$\boxed{b^+ = (A/D_0) e^{i\bar{\eta}},} \tag{C.28}$$

where the label $a$ is now dropped, due to the independence on $a$.

Again, from the Wronskian $W[\bar{u}_a, \bar{v}_a] = -k$, one finds

$$\boxed{\left(b^+ - a\right) D_a P_a^+ = -k.} \tag{C.29}$$

From Eqs. (C.26-C.29) one also finds the useful relations:

$$\frac{1}{b^+ - a} = \frac{1}{b_a - a} - \frac{i\,(D_a)^2}{k} = -\frac{D_a(P_a + iD_a)}{k}. \tag{C.30}$$

## C.3 Negative energy

For negative energies $E = -\frac{\hbar^2\kappa^2}{2\mu}$ obtained when $k$ is continued to imaginary values $i\kappa$, the quantity $b^+$ becomes real. For convenience, $b^+(i\kappa)$ is denoted as $\lambda(\kappa)$. One can see from Eq. (C.24) that for imaginary $k = i\kappa$, the irregular function $\bar{v}_a^+$ is exponentially decreasing at large distance. Equation (C.25) shows that if $\lambda(\kappa)$ happens to be equal to $a$, then $\bar{v}_a^+$ is proportional to the regular solution $\bar{u}_a$, as seen from Eq. (C.5). In this case, being both regular at the origin and at infinity, the solution corresponds to a bound state. This shows that $\lambda(\kappa)$ is simply the s-wave scattering length $a$ of the potential at which there is a bound state in the $\ell$th partial wave with energy $-\frac{\hbar^2\kappa^2}{2\mu}$.

## C.4 Calculation of the universal functions

### C.4.1 General case

The functions $\eta_0, \eta_\infty, D_0, A$ may in some cases be calculated analytically for a given tail of the potential $V$, for example in the case of a contact interaction (see below). If only the analytical forms of $f_0$ and $f_\infty$ are known at small distance, one may numerically integrate the radial Schrödinger equation with positive energy $E$ from the known $f_0$ and $f_\infty$ at small distance, outwards to large distances. This gives the long-range oscillations $(D_0)^{-1}\sin(kr+\eta_0-\ell\pi/2)$ and $(A)^{-1}\sin(kr+\eta_\infty-\ell\pi/2)$, from which $D_0, \eta_0, A$, and $\eta_\infty$ can be extracted.

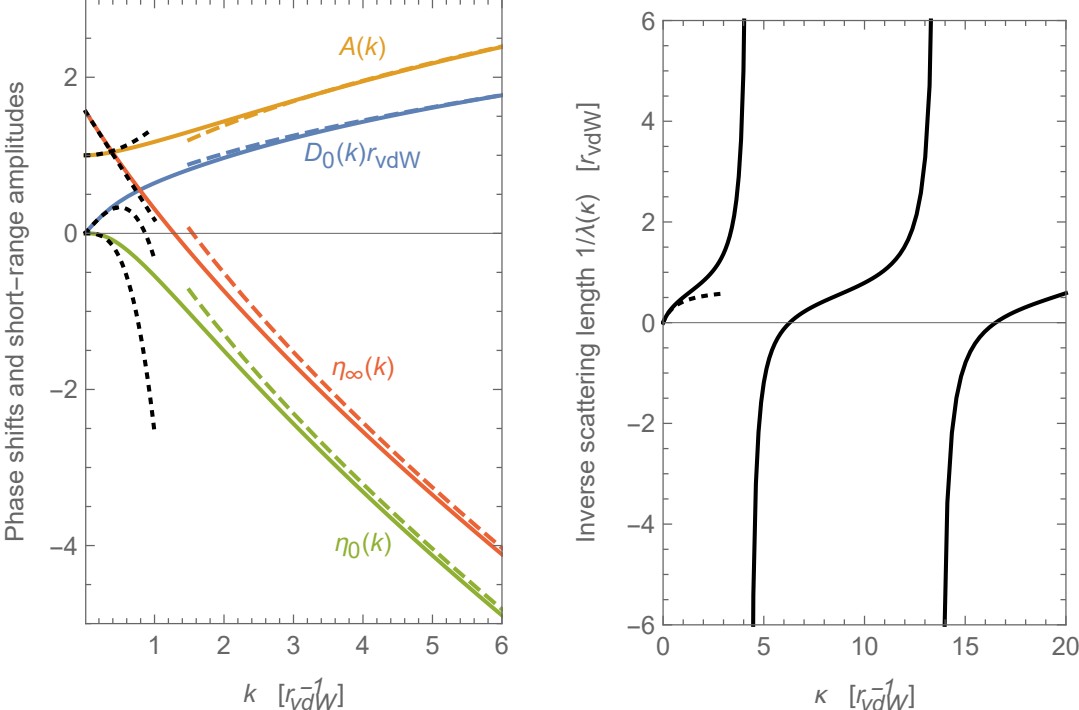

Figure 5: Universal functions in the s wave for potentials with van der Waals tail $V_{\text{tail}}(r) = -C_6/r^6$. Left panel: functions $\eta_0(k), \eta_\infty(k), D_0(k)$, and $A(k)$ for positive energies $\frac{\hbar^2 k^2}{2\mu}$. Right panel: function $1/\lambda(\kappa)$ for negative energies $-\frac{\hbar^2 \kappa^2}{2\mu}$. The quantity $\lambda(\kappa)$ is simply the scattering length for which the potential admits a bound state with binding energy $\frac{\hbar^2 \kappa^2}{2\mu}$. All quantities are plotted in units of the van der Waals length $r_{\text{vdW}} = \frac{1}{2}\left(2\mu C_6/\hbar^2\right)^{1/4}$. The dotted curves correspond to the small-$k$ formulas Eqs. (C.31-C.34) and Eq. (C.37) and the dashed curves correspond the large-$k$ formulas Eqs. (C.38-C.41).

To calculate $\lambda(\kappa)$, one can start at large distance from the exponentially decaying form $\exp(-\kappa r)$ and integrate inwards with negative energy $E = -\hbar^2 \kappa^2/2\mu$ to find the short-distance oscillations $f_0 - \lambda(\kappa) f_\infty$ and extract $\lambda(\kappa)$. Alternatively, one can calculate the bound state spectrum of the potential $V$ in the $\ell$th wave for different values of the scattering length $a$ set by altering the short-range part of $V$. In all cases, the universal functions $\eta_0, \eta_\infty, D_0, A$, and $\lambda(\kappa)$ can be easily obtained with these numerical procedures.

### C.4.2 Case of van der Waals interactions

For potentials with a van der Waals tail $-C_6/r^6$, the characteristic length scale is the van der Waals length $r_{\text{vdW}} = \frac{1}{2}\left(2\mu C_6/\hbar^2\right)^{1/4}$ or equivalently the mean scattering length $\bar{a} = \frac{4\pi}{\Gamma(1/4)^2} r_{\text{vdW}}$. One can in principle obtain analytical expressions of the universal functions from the analytical solution of the Schrödinger equation for van der Waals potentials [35], although they are rather involved. Alternatively, one can employ the numerical method sketched above. Figure 5 shows the result for the s wave.

The functions for the s wave admit the following analytical expressions for small $k \ll \bar{a}^{-1}$:

$$\eta_0(k) = -\frac{8}{3}\bar{a}r_{\mathrm{vdW}}k^3 + O(k^4), \tag{C.31}$$

$$\eta_\infty(k) = \frac{\pi}{2} - \frac{4}{3}\frac{r_{\mathrm{vdW}}^2}{\bar{a}}k + O(k^3), \tag{C.32}$$

$$D_0(k) = k - \frac{4}{3}r_{\mathrm{vdW}}^2 k^3 + O(k^4), \tag{C.33}$$

$$A(k) = 1 + \frac{4}{3}\left(1 - \frac{2}{3}\frac{r_{\mathrm{vdW}}^2}{\bar{a}^2}\right)k^2 r_{\mathrm{vdW}}^2 + O(k^3). \tag{C.34}$$

From this and using Eq. (C.10), one can perform the effective range expansion:

$$k \cot\eta_a = -\frac{1}{a} + \frac{1}{2}r_{\mathrm{eff}}k^2 + O\left(k^3\right), \tag{C.35}$$

yielding the effective range

$$r_{\mathrm{eff}} = r_{\mathrm{eff}}^{(\infty)}\left[\left(\frac{\bar{a}}{a}\right)^2 + \left(\frac{\bar{a}}{a} - 1\right)^2\right], \tag{C.36}$$

with $r_{\mathrm{eff}}^{(\infty)} = \frac{8}{3}\frac{r_{\mathrm{vdW}}^2}{\bar{a}}$.

For negative energies, one finds for small $\kappa \ll \bar{a}^{-1}$:

$$\lambda(\kappa) = \frac{1}{\kappa} + \frac{1}{2}r_{\mathrm{eff}}^{(\infty)} + O(\kappa). \tag{C.37}$$

One can also derive the following expressions in the high-energy limit $k \gg \bar{a}^{-1}$ using the WKB approximation:

$$\eta_0(k) = -\xi \times (kr_{\mathrm{vdW}})^{2/3} + 5\pi/8, \tag{C.38}$$

$$\eta_\infty(k) = -\xi \times (kr_{\mathrm{vdW}})^{2/3} + 7\pi/8, \tag{C.39}$$

$$D_0(k) = \sqrt{\frac{k}{2\bar{a}}}, \tag{C.40}$$

$$A(k) = \sqrt{k\bar{a}}, \tag{C.41}$$

with $\xi = -2^{-1/3}\pi^{-1/2}\Gamma\left(-\frac{1}{3}\right)\Gamma\left(\frac{5}{6}\right) \approx 2.0533$.

### C.4.3 Case of contact interactions

The case of contact interactions can be obtained by taking the limit $r_{\mathrm{vdW}} \to 0$ of Eqs. (C.31-C.34) and (C.37), yielding:

$$\eta_0(k) = 0, \tag{C.42}$$

$$\eta_\infty(k) = \frac{\pi}{2}, \tag{C.43}$$

$$D_0(k) = k, \tag{C.44}$$

$$A(k) = 1, \tag{C.45}$$

$$\lambda(\kappa) = 1/\kappa. \tag{C.46}$$

## C.5  Connection with other QDT notations

In the works of Refs. [15, 31, 32], the quantum defect theory is formulated with a set of four functions $Z_{ff}, Z_{gg}, Z_{fg}$, and $Z_{gf}$ along with a short-distance $K_0^0$ that is related to the scattering length $a$, such that the scattering phase shift reads:

$$\tan \eta_a = \frac{K_0^0 Z_{gg} - Z_{fg}}{Z_{ff} - K_0^0 Z_{gf}}, \quad \text{with} \quad K_0^0 = \left(1 - \frac{a}{\bar{a}}\right)^{-1}. \tag{C.47}$$

Therefore, the functions $\eta_0, \eta_\infty, D_0$, and $A$ are related to these functions by the relations:

$$\eta_0 = \arctan \frac{Z_{gg} - Z_{fg}}{Z_{ff} - Z_{gf}}, \tag{C.48}$$

$$\eta_\infty = -\arctan \frac{Z_{fg}}{Z_{ff}}, \tag{C.49}$$

$$(D_0)^{-1} = \bar{a} \sqrt{\left(Z_{gg} - Z_{fg}\right)^2 + \left(Z_{ff} - Z_{gf}\right)^2}, \tag{C.50}$$

$$(A)^{-1} = \sqrt{Z_{ff}^2 + Z_{fg}^2}, \tag{C.51}$$

and conversely,

$$Z_{gg} = \bar{a}^{-1} (D_0)^{-1} \sin \eta_0 - (A)^{-1} \sin \eta_\infty, \tag{C.52}$$

$$Z_{fg} = -(A)^{-1} \sin \eta_\infty, \tag{C.53}$$

$$Z_{gf} = (A)^{-1} \cos \eta_\infty - \bar{a}^{-1} (D_0)^{-1} \cos \eta_0, \tag{C.54}$$

$$Z_{ff} = (A)^{-1} \cos \eta_\infty. \tag{C.55}$$

Note that the four functions $\eta_0, \eta_\infty, D_0$, and $A$ shown in Fig. 5 all have a simple monotonic variation with $k$, whereas the four functions $Z_{ff}, Z_{gg}, Z_{fg}$, and $Z_{gf}$ have oscillatory variations.

In Refs [21, 33, 55], the short-distance energy-independent radial functions were connected to the long-range energy-normalised radial functions through a phase shift $\varphi$ and two amplitudes $A_k^{-1/2}$ and $\mathcal{G}_k$, also denoted as $C(E)$ and $\tan \lambda(E)$. With the current notations, $\varphi$ is the quantum defect related to $a$ through:

$$\tan \varphi = K_0^0 = \left(1 - \frac{a}{\bar{a}}\right)^{-1}, \tag{C.56}$$

and $C_k$ and $\mathcal{G}_k$ are related to $D_a, P_a$, and $b_a$ by the relations:

$$D_a(k) = (C_k)^{-1} \sqrt{\frac{k/\bar{a}}{1 + (1 - a/\bar{a})^2}}, \tag{C.57}$$

$$P_a(k) = -\sqrt{\frac{k/\bar{a}}{1 + (1 - r_0)^2}} \left(\mathcal{G}_k + 1 - \frac{a}{\bar{a}}\right) C_k, \tag{C.58}$$

$$b_a(k) = \bar{a} \frac{\frac{a}{\bar{a}}(\mathcal{G}_k - 1) + 2}{\mathcal{G}_k + 1 - \frac{a}{\bar{a}}}. \tag{C.59}$$

# D  Renormalised quantum defect theory of the isolated resonance

## D.1  Width and shift

Combining the results of the two preceding sections, one can now formulate the quantum defect theory of the isolated resonance. According to Eqs. (B.7), (B.20) and (B.12), the complex

shift $\Delta^+$ is given by

$$\Delta^+ = -\frac{2\mu}{\hbar^2 k}\int_0^\infty dr \int_0^\infty dr' w(r)\bar{u}_{\mathrm{o}}(r_<)\bar{v}_{\mathrm{o}}^+(r_>)w(r'). \tag{D.1}$$

Now, assuming that the coupling $w(r)$ is localised in the region $r_0 \lesssim r \ll r_{\mathrm{tail}}$ where Eqs. (C.7-C.25) can be used, and using Eq. (C.29) one finds

$$\Delta^+ = \frac{2\mu}{\hbar^2}\frac{(X - a_{\mathrm{o}}Y)(X - b^+Y) - (b^+ - a_{\mathrm{o}})Z}{b^+ - a_{\mathrm{o}}}, \tag{D.2}$$

with

$$X \equiv \int_0^\infty dr\, w(r)f_0(r), \tag{D.3}$$

$$Y \equiv \int_0^\infty dr\, w(r)f_\infty(r), \tag{D.4}$$

$$Z \equiv \int_0^\infty dr \int_r^\infty dr' w(r)w(r')\big(f_0(r)f_\infty(r') - f_\infty(r)f_0(r')\big). \tag{D.5}$$

Thus, introducing the lengths

$$a_{\mathrm{c}} \equiv X/Y, \tag{D.6}$$

$$a_{\mathrm{c}}' \equiv Z/Y^2, \tag{D.7}$$

one obtains

$$\boxed{\Delta^+ = \frac{\gamma}{b^+(k) - a_{\mathrm{o}}} + \Delta_0,} \tag{D.8}$$

with

$$\Delta_0 \equiv \frac{2\mu}{\hbar^2}Y^2\left(a_{\mathrm{o}} - a_{\mathrm{c}} - a_{\mathrm{c}}'\right), \tag{D.9}$$

$$\gamma \equiv \frac{2\mu}{\hbar^2}Y^2|a_{\mathrm{c}} - a_{\mathrm{o}}|^2 = \frac{2\mu}{4\pi\hbar^2}W_0^2\left(1 - \frac{a_{\mathrm{o}}}{a_{\mathrm{c}}}\right), \tag{D.10}$$

where the quantity $W_0 \equiv \sqrt{4\pi}(w|f_0)$ characterises the strength of the coupling between the open and closed channels. Note that $\Delta_0 = \lim_{k\to 0}\Delta$ in the case of the s wave ($\ell = 0$), for which $b^+(k) \xrightarrow[k\to 0]{} \infty$.

The simplicity of Eq. (D.8) is striking, as the dependence on the closed-channel parameters $W_0$, $a_{\mathrm{c}}$, and $a_{\mathrm{c}}'$ is entirely encapsulated in $\Delta_0$ and $\gamma$, while the dependence on the open-channel parameters only appears through the scattering length $a_{\mathrm{o}}$ in the denominator of Eq. (D.8). For energies $E$ below the open-channel threshold $E_{\mathrm{o}}$, the shift $\Delta^+$ and the length $b^+(ik) = \lambda(\kappa)$ are real, leading to the simple result:

$$\Delta = \frac{\gamma}{\lambda(\kappa) - a_{\mathrm{o}}} + \Delta_0. \tag{D.11}$$

For energies $E$ above the open-channel threshold $E_{\mathrm{o}}$, the real and imaginary parts of $\Delta^+ = \Delta - i\Gamma/2$ can be obtained from Eq. (D.8) using Eq. (C.30):

$$\Delta = \frac{\gamma}{b_{\mathrm{o}} - a_{\mathrm{o}}} + \Delta_0 = -\gamma\frac{D_{\mathrm{o}}P_{\mathrm{o}}}{k} + \Delta_0, \tag{D.12}$$

$$\frac{\Gamma}{2} = \gamma\frac{(D_{\mathrm{o}})^2}{k}. \tag{D.13}$$

Using the expressions Eqs. (C.20-C.22) one finds

$$\Delta = \Delta_0 + \frac{\Gamma}{2} B_o \, , \tag{D.14}$$

$$\frac{\Gamma}{2} = \gamma \frac{k}{(A)^2 \left[ 1 + (B_o)^2 \right]} \, . \tag{D.15}$$

## D.2  Low energy

In the low-energy limit, the general effective-range expansion of a resonance is given by Eq. (B.40).

In the case of a resonance with van der Waals interaction, it can be found from Eqs. (C.31-C.34) that

$$\alpha = \gamma \left( \frac{1}{2} r_{\text{eff}}^{(\infty)} - a_o \right) , \tag{D.16}$$

$$\beta = r_{\text{eff}}^{(\infty)} (a_o - \bar{a}) - a_o^2 \, . \tag{D.17}$$

In the isolated resonance limit where non-resonant contributions are negligible (i.e. $\tilde{\gamma} = \gamma$, $\tilde{\beta} = \beta$, and $a_o = a_{\text{bg}}$), this leads to the effective range,

$$r_{\text{eff}} = \left( r_{\text{eff}}^{(\infty)} - 2R_\star \right) \left( 1 - \frac{a_{\text{bg}}}{a} \right)^2 + r_{\text{bg}} \frac{a_{\text{bg}}^2}{a^2} + 2 r_{\text{eff}}^{(\infty)} \frac{a_{\text{bg}}}{a} \left( 1 - \frac{\bar{a}}{a_{\text{bg}}} \right) \left( 1 - \frac{a_{\text{bg}}}{a} \right) , \tag{D.18}$$

where $r_{\text{eff}}^{(\infty)} = \frac{8}{3} \frac{r_{\text{vdW}}^2}{\bar{a}}$ and $r_{\text{bg}}$ is the open-channel effective range given in terms of $a_{\text{bg}}$ by Eq. (C.36). It follows that close to the resonance ($a \to \infty$), the effective range reduces to:

$$\boxed{r_{\text{eff}} = r_{\text{eff}}^{(\infty)} - 2R_\star \, .} \tag{D.19}$$

One can see from this formula that there are two opposite limits: when $R_\star \ll r_{\text{eff}}^{(\infty)} \sim r_{\text{vdW}}$ (open-channel dominated resonance, a.k.a "broad" resonance [3]) the effective range $r_{\text{eff}} \approx r_{\text{eff}}^{(\infty)}$ is positive and approaches the effective range of the single-channel van der Waals potential at unitarity (see Eq. (C.36)), whereas when $R_\star \gg r_{\text{eff}}^{(\infty)} \sim r_{\text{vdW}}$ (closed-channel dominated resonance, a.k.a "narrow" resonance [3]), the effective-range is $r_{\text{eff}} \approx -2R_\star$ is negative and approaches the effective range of the zero-range two-channel model at unitarity [37].

Indeed, in the case of a resonance with contact interactions, it can be found from Eqs. (C.42-C.45), or simply by taking the limit $r_{\text{vdW}} \to 0$ in Eq. (D.18), that

$$r_{\text{eff}} = -2R_\star \left( 1 - \frac{a_{\text{bg}}}{a} \right)^2 , \tag{D.20}$$

which shows that the contact (zero-range) two-channel model has a negative effective range and thus always describes a closed-channel dominated Feshbach resonance.

## D.3  Short-distance amplitudes

The QDT gives a simple account of the wave function inside the tail region. The radial wave function in the open-channel component is given by the isolated resonance theory equation (B.25). Assuming that the coupling $w(r)$ is localised around a distance $r_w$, one can use Eqs. (B.19) and (B.30) to obtain the radial wave function for $r \gg r_w$:

$$u_o(r) \underset{r \gg r_w}{=} \bar{u}_o(r) + \tan(\eta - \eta_o) \bar{v}_o(r) \, . \tag{D.21}$$

This shows that for distances beyond the coupling region, the wave function is proportional to the solution of the open-channel potential with a short-distance boundary condition yielding the modified scattering phase shift $\eta$ instead of the original phase shift $\eta_o$.

One can also use Eqs. (B.25) and (B.19) to obtain the radial wave function for $r \ll r_w$:

$$u_o(r) \underset{r \ll r_w}{=} \left[ 1 - \left( \zeta_{nr} + \frac{\tilde{\Gamma}/2}{E - E_b - \Delta} \frac{(\bar{v}_o | w)}{(\bar{u}_o | w)} \right) \right] \times \bar{u}_o(r), \tag{D.22}$$

with

$$\zeta_{nr} \equiv \frac{2\mu}{\hbar^2 k} (\bar{v}_o | w_{nr}). \tag{D.23}$$

This shows that for distances beneath the coupling region the wave function is proportional to the unperturbed solution $\bar{u}_o$ of the open-channel potential $V_o$.

Now, assuming that the coupling region $r \sim r_w$ lies in the range $r_0 \lesssim r \ll r_{tail}$ where the wave functions $\bar{u}_o$ and $\bar{v}_o$ are energy independent, one can use the QDT formalism, namely Eqs. (C.7-C.8), to further specify the form of the radial wave function $u_o$. For $r \gg r_w$, one finds that $u_o$ is proportional to the zero-energy solution with an energy-dependent scattering length $a_{eff}$:

$$\boxed{u_o(r) \underset{r_w \ll r \ll r_{tail}}{=} \mathcal{A}_>(k) \times (f_0(r) - a_{eff}(k) f_\infty(r)),} \tag{D.24}$$

with the amplitude $\mathcal{A}_>$ and scattering length $a_{eff}$ given by:

$$\mathcal{A}_>(k) \equiv D_o + \tan(\eta - \eta_o) P_o, \tag{D.25}$$

$$a_{eff}(k) \equiv a_o - \frac{\tan(\eta - \eta_o) \frac{k}{D_o^2}}{1 + \tan(\eta - \eta_o) \frac{P_o}{D_o}}. \tag{D.26}$$

For $r \ll r_w$, the wave function is proportional to the zero-energy solution with the unperturbed scattering length $a_o$:

$$\boxed{u_o(r) \underset{r_0 \lesssim r \ll r_w}{=} \mathcal{A}_<(k) \times (f_0(r) - a_o f_\infty(r)),} \tag{D.27}$$

with the amplitude $\mathcal{A}_<$ given by:

$$\mathcal{A}_<(k) \equiv D_o - P_o \left( \xi_{nr} \frac{b_o - a_c^{nr}}{a_o - a_c^{nr}} + \frac{\tilde{\Gamma}/2}{E - E_b - \Delta} \frac{b_o - a_c}{a_o - a_c} \right), \tag{D.28}$$

where the non-resonant closed-channel scattering length $a_c^{nr}$ is defined by:

$$a_c^{nr} \equiv \frac{(f_0 | w_{nr})}{(f_\infty | w_{nr})}. \tag{}$$

In the s wave, for small $k$, one finds:

$$\mathcal{A}_>(k) \xrightarrow{k \to 0} k, \tag{D.29}$$

$$\mathcal{A}_<(k) \xrightarrow{k \to 0} \mathcal{A}_>(k) \frac{a - a_c - a_{nr} \frac{a_c - a_c^{nr}}{a_o - a_c^{nr}}}{a_o - a_c}, \tag{D.30}$$

$$a_{eff}(k) \xrightarrow{k \to 0} a. \tag{D.31}$$

In the fully isolated resonance limit where the non-resonant parts are negligible, the expressions of Eqs. (D.25, D.26, D.28) simplify to:

$$\mathcal{A}_>(k) = D_\text{o}(k) \frac{a_\text{o} - b_\text{o}(k)}{a_\text{eff}(k) - b_\text{o}(k)}, \tag{D.32}$$

$$\mathcal{A}_<(k) = \mathcal{A}_>(k) \frac{a_\text{eff}(k) - a_c}{a_\text{o} - a_c}, \tag{D.33}$$

$$a_\text{eff}(k) = a_\text{o} + \frac{\gamma}{E - E_\text{b} - \Delta_0}. \tag{D.34}$$

In particular, at low energy such that $a_\text{eff}(k) \approx a$, one finds the simple formula for the ratio:

$$\boxed{\frac{\mathcal{A}_<}{\mathcal{A}_>} = \frac{a - a_\text{c}}{a_\text{o} - a_\text{c}},} \tag{D.35}$$

showing that the short-distance amplitude vanishes when $a = a_\text{c}$. Note that the general formula Eq. (D.29) for a partially isolated resonance reduces to Eq. (D.35) in the special case where $a_\text{c}^\text{nr} = a_\text{c}$. This happens when the wave functions of the resonant and non-resonant bare states in the closed channel are also energy-independent in the coupling region, and thus characterised by the same scattering length.

## D.4 Application to lithium-6

The lithium-6 $ab$ diatomic resonance (where $ab$ designates the two lowest hyperfine states of lithium-6) near the magnetic field intensity $B = 834$ G is described by five hyperfine channels characterised by a total spin projection $m_F = 0$. The interaction between the atoms depends on the total electronic spin $S$ of the two valence electrons, which can be either in a singlet ($S = 0$) or triplet ($S = 1$) state. This multi-channel system with radial components $u_i(r)$ ($i = 1, \ldots, 5$) can thus be solved numerically using the singlet and triplet interaction potentials and the atomic hyperfine Hamiltonian.

The bare bound state causing this resonance has been identified as the $v = 38$ s-wave level of the singlet interaction potential, with radial wave function $u_\text{b}(r)$. Therefore, to construct the effective two-channel components, one can project the components $u_i$ onto the bare bound state to obtain the closed-channel component $u_\text{c}$, and project out the bare bound state and retain only the $ab$ entrance component ($i = 1$) to obtain the open-channel component $u_\text{o}$. Explicitly,

$$u_\text{c}(r) = \sqrt{\sum_{i,j=1}^{5} \left| \alpha_{i,j}(u_\text{b}|u_j) \right|^2} \, u_\text{b}(r), \tag{D.36}$$

$$u_\text{o}(r) = u_1(r) - \sum_{j=1}^{5} \alpha_{1j}(u_\text{b}|u_j) u_\text{b}(r), \tag{D.37}$$

where $\alpha_{ij}$ are the matrix elements of the projector $1 - \hat{S}^2$ onto the singlet state.

The zero-energy components are shown in Fig. 6 for two different values of the magnetic field intensity. The open-channel wave function $u_\text{o}$ (orange curve) is well fitted at large distance by the wave function of Eq. (D.24) (dashed curve), and at short distance by the wave function of Eq. (D.27) (dotted curve). The two fits deviate from $u_\text{o}$ in a region of distances around $r_\text{w} = 2.6$ nm, which shows that the inter-channel coupling is localised in that region. The QDT is therefore an accurate description for energies smaller than 240 $h$MHz $\sim 10$ mK

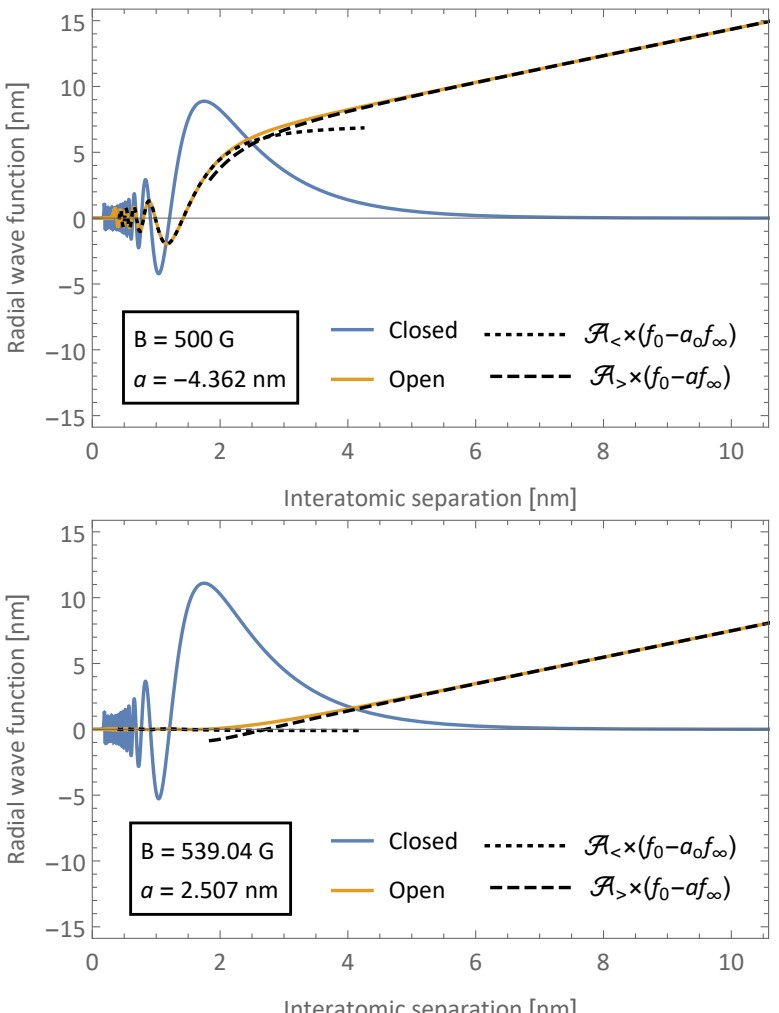

Figure 6: Effective two-channel radial wave functions $u_c$ ("closed", blue curve) and $u_o$ ("open", orange curve) of the lithium-6 diatomic $ab$ resonance near $B = 834$ G. Top: wave functions at $B = 500$ G, corresponding to a scattering length $a = -4.362$ nm. Bottom: wave functions at $B = 539.04$ G, corresponding to a scattering length $a = a_c = 2.507$ nm. The open-channel wave function $u_o$ is fitted at large distance by the wave function of Eq. (D.24) (dashed curve) and at short by the wave function of Eq. (D.27) (dotted curve).

above and below the threshold. The fits enable to extract the amplitudes $\mathcal{A}_>$ and $\mathcal{A}_<$, as well as the open-channel scattering length $a_o$ and the physical scattering length $a$.

Both scattering lengths $a_o$ and $a$ are plotted as a function of magnetic field intensity as blue and orange curves in Fig. 7. One can see that the open-channel scattering length $a_o$ is close to the triplet scattering length $a_t = -112.8$ nm, confirming the spin triplet character of the open channel, while the physical scattering length $a$ is well reproduced by the formula of Eq. (B.34). This yields the background scattering length $a_{bg}$, which has a small dependence on the magnetic field as shown by the green curve in Fig. 7. This dependence is captured by the following Taylor expansion around $B_0$:

$$a_{bg} = a_{bg}^{(0)} + a_{bg}^{(1)} (B/B_0 - 1) + a_{bg}^{(2)} (B/B_0 - 1)^2 , \qquad (D.38)$$

with $a_{bg}^{(0)} = -84.89$ nm, $a_{bg}^{(1)} = -24.19$ nm, and $a_{bg}^{(2)} = 22.77$ nm.

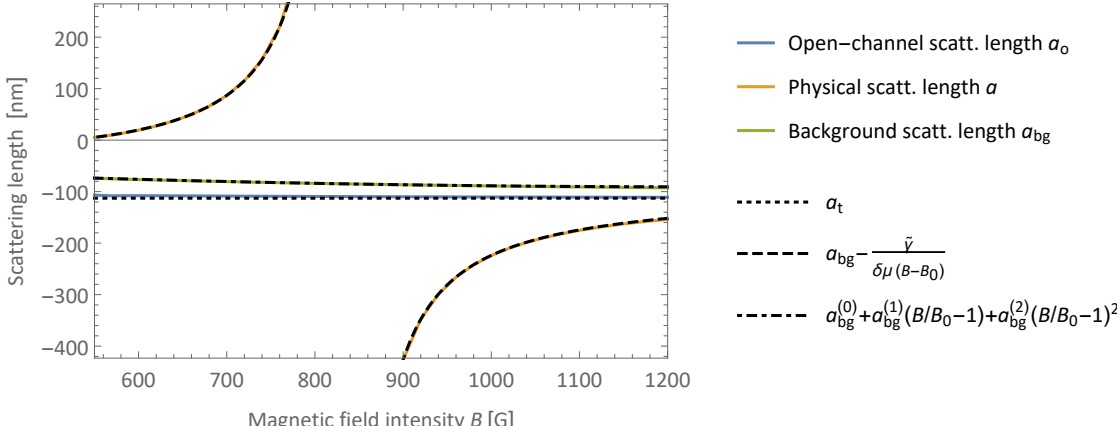

Figure 7: Open-channel scattering length $a_o$ (orange curve), physical scattering length $a$ (blue curve), and background scattering length $a_{bg}$ of the lithium-6 diatomic $ab$ resonance near $B = 834$ G, obtained by fitting the numerical wave functions, as a function of magnetic field intensity. The open-channel scattering length $a_o$ is close to the triplet scattering length $a_t = -112.8$ nm (dotted line), while the physical scattering length is well reproduced by Eq. (B.34) (dahsed curve) with $\tilde{\gamma}/h = 62\,770$ MHz nm, and $\tilde{E}_b = E_b + \Delta_0 - E_o = \delta\mu \times (B - B_0)$ with $\delta\mu/h = 2.8$ MHz/G and $B_0 = 834.08$ G. This yields the background scattering length $a_{bg}$, which is well reproduced by Eq. (D.38).

The ratio $|\mathcal{A}_< / \mathcal{A}_>|$ is plotted in Fig. 8 as a function of magnetic field intensity. It is well reproduced by the formula Eq. (D.35) with a closed-channel scattering length $a_c = 2.507$ nm $\approx a_s$ and an open-channel scattering length $a_o = -109.6$ nm $\approx a_t$. It should be noted that the non-resonant contribution is not negligible for this resonance: the background scattering length $a_{bg}$ is found to be around -85 nm, significantly differing from $a_o$. Thus, in principle one may not use Eq. (D.35) which is obtained in the fully isolated resonance limit, but Eq. (D.30), which includes the non-resonant correction. However, it turns out that $a_o$ is very large compared to $a_c$ and $a_c^{nr}$ (both are presumably of the same order, or even possibly equal) so that Eq. (D.35) is a good approximation of Eq. (D.30) in this case.

One can see that the short-distance amplitude vanishes at the magnetic field intensity $B = 539.04$ G corresponding as expected to $a = a_c$. This suppression of the open-channel amplitude at short distance can be visualised in the bottom panel of Fig. 6. A close look around this magnetic field (see the bottom panel of Fig. 8) reveals that a very narrow resonance accidentally occurs close to that point. Although the presence of this extra resonance, which is due to a bound state with total nuclear spin $I = 2$ [3], complicates a bit the variation of $|\mathcal{A}_< / \mathcal{A}_>|$, it is still reproduced by Eq. (D.35) with the same value of $a_c$ when the precise variation of $a$ (inluding the narrow resonance) is taken into account. This is because both the broad and narrow resonances originate from the same bound state $\nu = 38$ of the singlet potential, thus having the same values of $a_c$. The fact that the obtained value of $a_c = 2.507$ nm is very close to the singlet scattering length $a_s = 2.391$ nm confirms that the singlet character of the closed-channel bound state.

Since the open channel corresponds essentially to the triplet component, an experiment probing the triplet component, for instance by photoassociation, could reveal how the amplitude of the open-channel wave function vanishes near $a = a_c$. This is illustrated in Fig. 8,

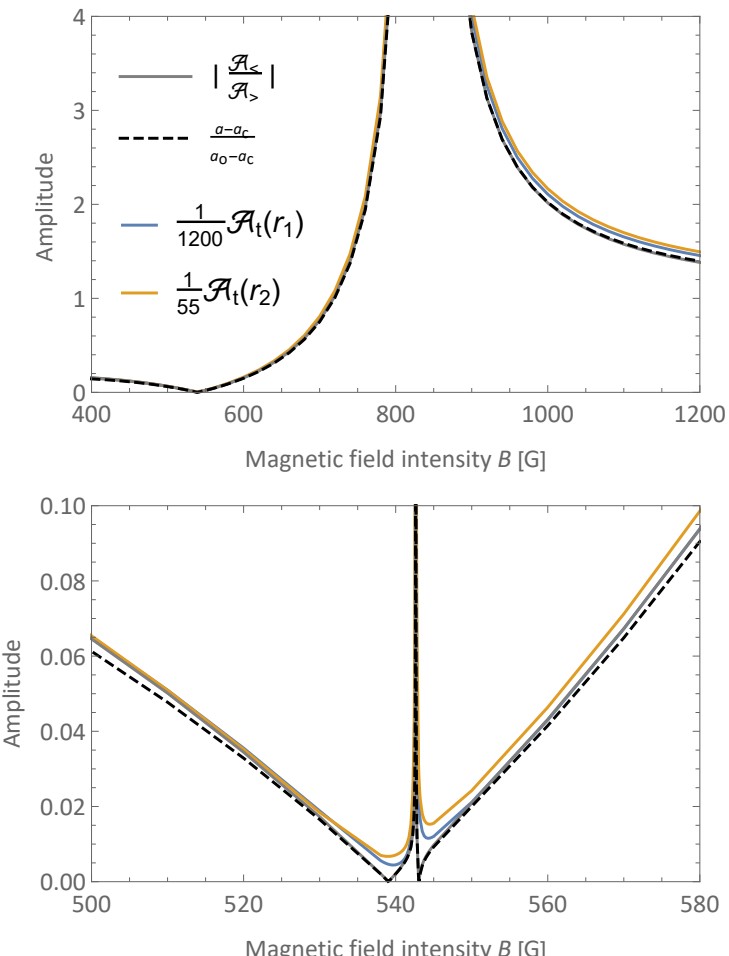

Figure 8: Top: Short-distance amplitude of the lithium-6 diatomic open channel as a function of magnetic field intensity. The grey curve shows the ratio of the amplitudes $\mathcal{A}_<$ and $\mathcal{A}_>$ of Eqs. (D.24-D.27). It is well reproduced by Eq. (D.35) with $a_c = 2.510$ nm (dashed curve). The blue and orange curves show the triplet amplitude $\mathcal{A}_t(r)$ of Eq. (D.39) at the probing distance $r_1 = 1.9$ nm (blue) and $r_2 = 0.8$ nm (orange). Bottom: closeup of the top figure in the region where the short-distance amplitude vanishes.

where the triplet amplitude,

$$\mathcal{A}_t(r) \equiv \sqrt{\sum_{i,j=1}^{5} \left| (1 - \alpha_{i,j}) u_j(r) \right|^2}, \tag{D.39}$$

at two different probing distances $r$ is plotted as a function of the magnetic field intensity. For both probing distances, the triplet amplitude reproduces very well the short-distance amplitude $\mathcal{A}_<$ up to a scaling factor. However, very close to the points where $\mathcal{A}_<$ vanishes, the triplet amplitude does not completely vanish, an indication that the triplet component does not perfectly account for the open channel, but also includes non-vanishing admixtures. Nevertheless, the measurement of $\mathcal{A}_<$ over a wide enough range of magnetic field intensities would enable to determine $a_c$.

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
