# Peer review of "Closed-channel parameters of Feshbach resonances"

_SciPost Physics, doi:SciPost Phys. 18, 036 (2025)_

## Round 2 · Referee Report · Anonymous (Referee 1) · 2024-8-26

Report

In this manuscript, the author uses a coupled-channel formalism to study the Feshbach resonance in a system with a single two-body bound state (called closed channel in the manuscript) coupled to a two-body continuum (called open channel in the manuscript). The main result is that if the potential in the open channel is deep enough at large distance, like the magnetic Feshbach resonances in the ultracold atoms, the quantum defect theory (QDT) can be applied and the two-body observable like the binding/resonance energy related to the Feshbach resonance is insensitive to the details of the bare bound state, i.e., insensitive to the parameters in the closed channel, which is different from the resonances involving shallow interaction potentials, such as hadron resonances. The author further points out that the three-body observables can be affected by the closed channel parameters such as the closed channel scattering length, which is suggested to be determined from the photoassociation experiments by the author. The manuscript renews our understanding and reveals some universality on the Feshbach resonance with deep open channel potential. Therefore, I recommend this manuscript for publication in SciPost Physics after the following questions have been addressed.

1) In Figs. 1 and 2, the author shows the dressed energy spectrum below the two-body threshold for a Feshbach bound state in a nonrelativistic model and in the QDT. How about the energy spectrum for a Feshbach resonance above the two-body threshold in these two models?

2) In Fig. 2, the dressed bound state energy of the Feshbach resonance has a big change near B0=202 G. The author should explain the physical mechanism for such a large change.

3) The author claims that the two-body observables of resonances involving
shallow interaction potentials like hadron resonances can be sensitive to the closed channel parameters. Are the parameters physical? Is it possible to extract some information about the bare resonance in the closed channel for these resonances?
In particular, the author suggests extracting the closed channel scattering length from the photoassociation experiment. Is it possible to extract the closed channel parameters for near-threshold hadron resonances in similar experiments?
For example, for the X(3872) resonance as a DˉD composite state with a cˉc core, how can the information about the core be obtained from experiments?

4) Some typos in the manuscript should be corrected.

In the seventh line in the caption of Fig. 2, Eq (7) -> Eq. (7).

In the third paragraph of Section 4.3, threhsold -> threshold.

Recommendation

Ask for minor revision

  • validity: -
  • significance: -
  • originality: -
  • clarity: -
  • formatting: -
  • grammar: -

Author:  Pascal Naidon  on 2024-10-22  [id 4887]

(in reply to Report 1 on 2024-08-26)

Attached below is a new version of the paper with modified parts highlighted in red.

Attachment:

FullPaper40-10.pdf

Author:  Pascal Naidon  on 2024-10-22  [id 4886]

(in reply to Report 1 on 2024-08-26)

I thank the referee for their positive assessment of the manuscript. Below are the replies to their questions.

** Question 1**

In Figs. 1 and 2, the author shows the dressed energy spectrum below the two-body threshold for a Feshbach bound state in a nonrelativistic model and in the QDT. How about the energy spectrum for a Feshbach resonance above the two-body threshold in these two models?

Above the two-body threshold, the energy spectrum is a continuum characterised by scattering phase shifts. It features a resonant state with a formally complex energy. The real part of the energy is given by the bare state energy shifted by the real part of the shift given by Eq. (1). It corresponds to the energies where the sine squared of the scattering phase shift reaches unity. A white curve has been added to Fig. 2 (van der Waals QDT model) to show the position of this resonance. As to Fig. 1 (non-relativistic Gaussian model), it has been divided into three panels to show the scattering continuum in a similar way to Fig. 2.

** Question 2**

In Fig. 2, the dressed bound state energy of the Feshbach resonance has a big change near B0=202 G. The author should explain the physical mechanism for such a large change.

I suppose that the referee refers to the apparent large discontinuity around the resonance point B0=202G. This is because for this resonance there is a low-lying bound state in the open channel (whose energy is shown as a dotted line). Its coupling with the bare bound state (shown as one of the diagonal dashed lines) creates an avoided crossing, as discussed at the end of Section 5.1, resulting in two branches: one that connects to the threshold at B0=202G and asymptotes to the open-channel bound state energy on the left side, and another branch that asymptotes to the open-channel bound state energy on the right side. A sentence has been added to the caption of Fig. 2 to explain this fact. I hope this answers the referee’s question.

** Question 3**

The author claims that the two-body observables of resonances involving shallow interaction potentials like hadron resonances can be sensitive to the closed channel parameters. Are the parameters physical? Is it possible to extract some information about the bare resonance in the closed channel for these resonances? In particular, the author suggests extracting the closed channel scattering length from the photoassociation experiment. Is it possible to extract the closed channel parameters for near-threshold hadron resonances in similar experiments? For example, for the X(3872) resonance as a D¯D∗ composite state with a c¯c core, how can the information about the core be obtained from experiments?

I thank the referee for this very relevant question. Yes, it is in principle possible to extract some information about the compact core causing the resonance in X(3872). A discussion has been added at the end of Section 4.3 to show how the simple non-relativistic Gaussian model (which is not in the QDT regime) can assign a definite value to the mass of this compact core from two-body observables. This model is of course unrealistic, and the obtained value is at best very inaccurate. Nevertheless, it illustrates how a physical value could be ultimately obtained from two-body observables using a more realistic model. It is beyond the purpose of the present work to devise such a realistic model. As to the analogue of the photoassociation experiment for hadronic systems, it would also be possible in principle using some three-body process. However, such discussions are also beyond the scope of this work, since the parameters could in principle be determined from simpler two-body processes.

** Question 4 **

Some typos in the manuscript should be corrected. In the seventh line in the caption of Fig. 2, Eq (7) -> Eq. (7). In the third paragraph of Section 4.3, threhsold -> threshold.

I thank the referee for spotting these typos, which have been corrected.

Author:  Pascal Naidon  on 2024-10-22  [id 4885]

(in reply to Report 1 on 2024-08-26)
Category:
answer to question

I thank the referee for their positive assessment of the manuscript. Below are the replies to their questions.

1) In Figs. 1 and 2, the author shows the dressed energy spectrum below the two-body threshold for a Feshbach bound state in a nonrelativistic model and in the QDT. How about the energy spectrum for a Feshbach resonance above the two-body threshold in these two models?

---

## Round 3 · Referee Report · Anonymous (Referee 1) · 2024-11-1

Report

The author has answered all my comments properly. I would like to recommend accepting the current version for publication in SciPost.

Recommendation

Publish (meets expectations and criteria for this Journal)

---

## Round 3 · Referee Report · Anonymous (Referee 2) · 2024-11-28

Report

In the present work, the author shows that observations of a Feshbach resonance may not uniquely define the properties of the associated closed channel, especially in the case where the Feshbach resonance can be well-described by quantum defect theory (QDT). This paper builds upon the author's previous work on the properties of the closed channel associated with a Feshbach resonance, deriving the closed-channel scattering length in Ref. 21.

This result appears sound to me: the author cites several papers which successfully employ QDT to describe a Feshbach resonance while remaining deliberately ignorant of properties of the closed channel.

The author also demonstrates that the short-distance physics (which QDT hides) shows some dependency on the closed channel parameters, and that these may be studied with photoassociation spectroscopy.

Most of the evidence for the author's claims is relegated to the extensive appendices.

I thank the author for a well-written, stimulating manuscript, and recommend its publication after correcting the following typos:

- Immediately before Eq. 1, Appendix A and B -> Appendices A and B
- There are en dashes ("...grey - we come back..." on pg. 3 and "...negative value - (see Appendix..." on pg. 5) which should be em dashes
- Superfluous quotation mark in Fig 6 caption when referring to the orange curve

Recommendation

Ask for minor revision

---

## Round 3 · List of Changes

Following the referee's questions,

* Fig. 1 has been divided into three panels to show the scattering continuum in a similar way to Fig. 2. A white curve has been added in both Fig. 1 and 2 to show the location of the resonance in the continuum.
* A discussion has been added at the end of Section 4.3 to show how a model outside the QDT regime can assign a mass to the compact core responsible for a hadron resonance such as X(3872).
* Accordingly, sections have been added/expanded in the appendices to include a treatment of the effective range .

All changes can be seen in the file attached in the reply to the Referee.

---

## Round 4 · Author Response

Dear Editor,
Please find the resubmission of my manuscript.
Best regards,
Pascal Naidon

---

## Round 4 · List of Changes

All typos pointed out by the referee have been corrected.

---

## Editorial Decision

published